# Global biosphere-–climate interaction: a multi-temporal scale appraisal of observations and models

Jeroen Claessen[1], Annalisa Molini[2], Brecht Martens[1], Matteo Detto[3], Matthias Demuzere[1,4], and Diego G. Miralles[1]

[1]Laboratory of Hydrology and Water Management, Department of Environment, Ghent University, Ghent, Belgium
[2]Masdar Institute, Khalifa University of Science and Technology, Abu Dhabi, United Arab Emirates
[3]Department of Ecology and Evolutionary Biology, Princeton University, Princeton, United States of America
[4]Department of Geography, Ruhr-University Bochum, Bochum, Germany
**Correspondence:** Jeroen Claessen (Jeroen.Claessen@UGent.be)

**Abstract.** Improving the skill of Earth System Models (ESMs) in representing climate–vegetation interactions is crucial to enhance our predictions of future climate and ecosystem functioning. Therefore, ESMs need to correctly simulate the impact of climate on vegetation, but likewise, feedbacks of vegetation on climate must be adequately represented. However, model predictions at large spatial scales remain subjected to large uncertainties, mostly due to the lack of observational patterns to benchmark them. Here, the bi-directional nature of climate–vegetation interactions is explored across multiple temporal scales by adopting a spectral Granger causality framework that allows identifying potentially co-dependent variables. Results based on global and multi-decadal records of remotely-sensed leaf area index (LAI) and observed atmospheric data show that the climate control on vegetation variability increases with longer temporal scales, being higher at inter-annual than multi-month scales. Globally, precipitation is the most dominant driver of vegetation at monthly scales, particularly in (semi-)arid regions. The seasonal LAI variability in energy-driven latitudes is mainly controlled by radiation, while air temperature controls vegetation growth and decay in high northern latitudes at inter-annual scales. These observational results are used as benchmark to evaluate four ESM simulations from the Coupled Model Intercomparison Project Phase 5 (CMIP5). Findings indicate a tendency of ESMs to over-represent the climate control on LAI dynamics, and a particular overestimation of the dominance of precipitation in arid and semi-arid regions at inter-annual scales. Analogously, CMIP5 models overestimate the control of air temperature on seasonal vegetation variability, especially in forested regions. Overall, climate impacts on LAI are found to be stronger than the feedbacks of LAI on climate in both observations and models; in other words, local climate variability leaves a larger imprint on temporal LAI dynamics than vice versa. Note however that while vegetation reacts directly to its local climate conditions, the spatially collocated character of the analysis does not allow for the identification of remote feedbacks, which might result in an underestimation of the biophysical effects of vegetation on climate. Nonetheless, the widespread effect of LAI variability on radiation, as observed over the northern latitudes due to albedo changes, is overestimated by the CMIP5 models. Overall, our experiments emphasise the potential of benchmarking the representation of particular interactions in online ESMs using causal statistics in combination with observational data, as opposed to the more conventional evaluation of the magnitude and dynamics of individual variables.

## 1 Introduction

The biosphere is a key actor in the global carbon and water cycles, mainly through its impact on the energy balance at the Earth's surface and the chemistry of the atmosphere (McPherson, 2007; Pearson et al., 2013; Le Quéré et al., 2018). Long-
term patterns in temperature, incoming radiation and water availability strongly control the global distribution of biomes, while vegetation in turn alters climate via a series of local and remote feedbacks (Kottek et al., 2006; Bonan, 2008). In boreal regions, for example, vegetation is thought to preferentially warm the atmosphere (positive feedback) by lowering the surface albedo, while in tropical regions, it is thought to have a local net cooling effect (negative feedback), mainly due to high transpiration (Bonan, 2008; Forzieri et al., 2017). In fact, a net warming effect has been reported after tropical deforestation and agricultural
expansion (Alkama and Cescatti, 2016; Duveiller et al., 2018b). Furthermore, the biosphere also provides a negative climate feedback by acting as a net carbon sink (Schimel et al., 2015). This strong regulating power of vegetation in the Earth system indicates the need to accurately incorporate biosphere–climate interactions in the models used to predict changes in terrestrial ecosystems and future climate (Piao et al., 2013; Pachauri et al., 2014; Le Quéré et al., 2018). The different approaches to objectively evaluate the skill of Earth System Models (ESMs) in representing the two-way coupling between vegetation and
climate have revealed several model limitations (Randerson et al., 2009; Weiss et al., 2012; Murray-Tortarolo et al., 2013; Alessandri et al., 2017; Green et al., 2017; Duveiller et al., 2018a; Forzieri et al., 2018). Most of these efforts focus on the evaluation of the magnitude and short-term dynamics of individual variables (such as LAI, and gross primary production, GPP), rather than on the inter-variable sensitivities, which would be more informative on whether the interplay between vegetation and climate is reliably represented in these models. Furthermore, previous benchmark studies have typically focused on one
specific time scale (typically annually or monthly), while the ecosystem response to (and feedback on) climate is expected to vary for different time scales; e.g. a model may accurately replicate the observed interplay between vegetation and climate at monthly scales, but still fail to capture the sensitivities that become relevant at seasonal or inter-annual time scales.

Nonetheless, a first and necessary requirement towards improving the predictive skill of ESMs is the availability of data that can be used as reference. Satellite observations of our biosphere, hydrosphere and atmosphere are now widely available,
providing multi-decadal records of climatological and environmental variables at global scale that can be used as benchmark. Several studies have already focused on identifying short- and long-term global impacts of climate on vegetation using observational data, mostly from satellites (Nemani et al., 2003; Zhao and Running, 2010; Forkel et al., 2014; De Keersmaecker et al., 2015; Wu et al., 2015; Seddon et al., 2016; Papagiannopoulou et al., 2017b). Likewise, observational data have been used to benchmark vegetation variability in ESM's (Anav et al., 2013; Murray-Tortarolo et al., 2013), and an overestimation of mod-
elled annual LAI due to problems related to the timing of the phenological cycle has been suggested (Forkel et al., 2014; Verger et al., 2016). Rather than using correlation or regression techniques to address this issue, a method capable of inferring causality can greatly aid our understanding of key climate–biosphere processes, which in turn can help enhance the ESMs (Runge et al., 2019). In a recent example, Papagiannopoulou et al. (2017a, b) focused on evaluating multi-month vegetation variability

in response to local climate, using a non-linear Granger causality framework applied to optical remote sensing indices. They showed that water availability and precipitation patterns primarily drive vegetation anomalies at monthly scales in more than 60% of the vegetated land, but did not address the relevant drivers over longer time scales. The inter-annual variability in terrestrial carbon fluxes has also been intensively explored in recent years, with apparent contradictions in the findings regarding the importance of water availability and air temperature for biosphere dynamics (Jung et al., 2017; Humphrey et al., 2018; Green et al., 2019; Stocker et al., 2019). In addition, most studies to date have either attributed the covariance of vegetation and climate dynamics to the role of atmospheric processes driving biosphere variability (e.g., Nemani et al., 2003; Zhao and Running, 2010; Forkel et al., 2014; De Keersmaecker et al., 2015; Wu et al., 2015; Papagiannopoulou et al., 2017b), or to the opposite processes, i.e. the feedbacks of vegetation on climate (e.g., Forzieri et al., 2017; Zeng et al., 2017); to the authors knowledge, the study by Green et al. (2017) is the only exception in which the causal directionality of vegetation–climate interactions has been formally disentangled at global scales. In that study, a linear Granger causality approach was used to successfully unravel impacts and feedbacks between biosphere and climate at multi-month scales. However, the traditional Granger causality framework is unsuited to identify which interactions dominate at different temporal scales, thus to differentiate between the dominant causes and effects at multi-month, seasonal and inter-annual scales (Detto et al., 2012).

Here, we investigate climate–vegetation interactions over the global domain using an innovative variant of Granger causality, referred to as Conditional Spectral Granger Causality (CSGC) – see Dhamala et al. (2008) and Detto et al. (2012). CSGC relies on transforming time series from the time domain into a time–frequency space using the continuous wavelet transform, enabling the simultaneous analysis of interactions that are active at different temporal scales, from (e.g.) monthly to inter-annual. In addition, this technique allows for evaluating the contribution of any variable while conditioning on the others, and, because CSGC can cope with lagged responses, it enables the assessment of bi-directional interactions (Dhamala et al., 2008; Detto et al., 2012; see Sect. 2.2). The latter implies that the vegetation feedback on climate can be quantified separately from the climate impact on vegetation. In this study, CSGC is first applied to satellite observations to reveal useful insights regarding the global, multi-temporal scale, bi-directional interaction between vegetation dynamics and local climate (Sect. 3.1 and 3.3). Next, to benchmark the ESM representation of these biosphere–climate interactions, the approach is replicated using the outcome from four online simulations from the Coupled Model Intercomparison Project Phase 5 (CMIP5) models (Taylor et al., 2012; see Sect. 3.2 and 3.3). By comparing the observational and model-based results, areas with matching or diverging inter-variable sensitivities are identified.

## 2 Data and methods

### 2.1 Data

Multiple satellite-based data sets are used to evaluate the representation of climate–vegetation interactions in ESMs. The focus is on the key climatic drivers of vegetation growth, here assumed to be precipitation, net radiation and air temperature, consistent with previous studies (Nemani et al., 2003; Seddon et al., 2016; Jung et al., 2017; Papagiannopoulou et al., 2017b). Vegetation dynamics are diagnosed using LAI; in the following, when vegetation (state) is mentioned, the latter refers to LAI

unless stated otherwise. All datasets have global coverage, are processed into 0.5° spatial resolution via bilinear interpolation, and are averaged to monthly values prior to the application of CSGC.

### 2.1.1 Observational data

To avoid product-specific biases and artefacts, an ensemble of multiple observation-based products for each variable is created, consisting of: (a) four LAI, (b) two air temperature, (c) two net radiation, and (d) three precipitation data sets. The larger ensemble of data sets here adopted to characterise LAI and precipitation is motivated by the larger disparity among the different products of these variables (Jiang et al., 2017; Sun et al., 2018). LAI products have data gaps and higher uncertainties in winter periods (Yang et al., 2006; Xiao et al., 2016; Jiang et al., 2017). Gaps are here filled by bilinear interpolation, as CSGC requires continuous time series. Tab. 1 provides an overview of the available data sets resulting in the overlapping analysis period 1982–2015. The main observational results are based on the average of the 48-member ensemble, acquired by analysing all possible data set combinations. The effect of irrigation is quantified using the AQUASTAT Global Map of Irrigation Areas version 5.0, which provides the area equipped for irrigation expressed as percentage of the total area (Siebert et al., 2013). Finally, the International Geosphere–Biosphere Program (IGBP) land cover classification (Loveland and Belward, 1997) is used to determine biome-specific behaviours. At a biome-level, the mean observed and modelled interactions are calculated, and the range in ESM results is determined. These biomes include mixed forest (MF), deciduous broadleaf forest (DBF), deciduous needleleaf forest (DNF), evergreen broadleaf forest (EBF), evergreen needleleaf forest (ENF), barren or sparsely vegetated (BSV), cropland or natural vegetation mosaic (CNVM), cropland (C), grassland (G), savanna (S), woody savanna (WS), and open shrubland (OS).

### 2.1.2 Earth System Model data

A selection of coupled ESMs from the Coupled Model Intercomparison Project Phase 5 (CMIP5; Taylor et al., 2012) is assessed in their representation of climate–vegetation interactions. This includes the Hadley Global Environment Model 2 - Earth System (HadGEM2-ES; Collins et al., 2011), Institut Pierre Simon Laplace - Component Models 5 - Medium Resolution (IPSL-CM5A-MR; Dufresne et al., 2013), Norwegian Earth System Model 1 - Medium Resolution (NorESM1-M; Bentsen et al., 2013), and Community Climate System Model 4 (CCSM4; Gent et al., 2011). This selection is based on (a) use of similar land surface schemes as the Trends in Net Land-Atmosphere Exchange (TRENDY; Sitch et al., 2015) initiative, in order to allow for comparison with studies focusing on TRENDY models, (b) availability of hourly input data for air temperature, precipitation and net radiation (aggregated to monthly values in this study), and (c) model consideration of dynamic vegetation (Anav et al., 2015). Coupled model simulations are used to evaluate the full extent of vegetation feedbacks on climate. Using the historical input climate data, one realisation was used for each model to simulate vegetation dynamics, resulting in a monthly time series of LAI. Due to the discontinuation of historical simulations in 2005, the overlap with the observational record is limited to 24 complete years. To enhance the robustness of the results, the analysis period considers the entire 1956–2005 in the case of ESMs, under the assumption that the sensitivities are stationary (see e.g. Green et al., 2017). Sect. 3.2 addresses the validity of this assumption. Nonetheless, we acknowledge that the non-stationarity associated with changes in land use and

**Table 1.** Summary of global data sets used for vegetation, i.e. LAI, and climate, i.e. air temperature (Ta), net radiation (Rn), and precipitation (P).

| Product | Variable | Spatial resolution | Temporal resolution | Temporal coverage | Reference |
|---|---|---|---|---|---|
| Global Inventory Modelling and Mapping Studies 3rd generation (GIMMS3g) | LAI | $1/12°$ | Bimonthly | 1982–2015 | Zhu et al. (2013) |
| NOAA/AVHRR Thematic Climate Data Record (TCDR) Reflectance | LAI | $0.05°$ | Daily | 1982–2018 | Claverie et al. (2016) |
| GIMMS3g + Terra/MODIS C5 reflectance (GLOBMAP) | LAI | $1/13.75°$ | 28-day | 1982–2017 | Liu et al. (2012) |
| NOAA/AVHRR LTDR + Terra/MODIS C5 reflectance (GLASS) | LAI | $0.05°$ | 8-day | 1982–2015 | Xiao et al. (2016) |
| European Centre for Medium-Range Weather Forecasts (ECMWF) ERA5 | Ta, Rn and P | 32km | Hourly | 1979–present | Hersbach and Dee (2016) |
| Climate Research Unit - National Centers for Environmental Prediction (CRU-NCEP) version 7 | Ta, Rn and P | $0.05°$ | 6-hour | 1901–2016 | Viovy (2018) |
| Global Precipitation Climatology Centre (GPCC) | P | $0.5°$ | Daily | 1891–2016 | Schneider et al. (2011) |

land cover may induce divergences between the observation and model results. The latter will be presented as the average over the four model ensemble members.

## 2.2 Methods

Multi-temporal scale interactions between climate and vegetation are here explored using CSGC. To describe the method comprehensively, we first introduce the Granger causality in its classical formulation (parametric in the time domain; Sect 2.2.1), followed by the derivation of its spectral counterpart (non-parametric in the time-frequency domain; Sect. 2.2.2 and 2.2.3).

### 2.2.1 Granger Causality: Time domain formulation

According to Granger (1969), causality can be inferred if a predictor $X$ ($[x_1, x_2...x_{n-1}, x_n]$), with $n$ the number of time steps, contains information in past terms that aids the prediction of a target variable $Y$ ($[y_1, y_2...y_{n-1}, y_n]$), while this information is not contained in any other predictor or past values of the target variable itself. To assess the predictive power of $X$ on $Y$, the self-explanatory power of $Y$, i.e. the autocorrelation, has to be determined first, so it can later be factored out. At time $t$, the auto-predictive power of $Y$ can be calculated with the following univariate autoregressive equation:

$$y_t = \sum_{i=1}^{m} a_i y_{t-i} + \epsilon_t \tag{1}$$

where $m$ defines the maximum order of the autoregressive model (with $m \leq n$), $i$ is the time lag, $a_i$ are the coefficients describing the linear interaction between different time steps, and $\epsilon_t$ is the prediction error. Note that the order $m$ defines the maximum lag that is investigated, which does not necessarily imply that all predictors have an effect up to time step $m$. By increasing $m$, more lags are included, at the cost of increasing the computational demand.

The predictive power of $X$ on $Y$ can be assessed through construction of a second autoregressive model, containing a term capturing the contribution of $X$, given by:

$$y_t = \sum_{i=1}^{m} a_i y_{t-i} + \sum_{i=1}^{m} b_i x_{t-i} + \eta_t \tag{2}$$

with $\eta_t$ representing the prediction error of the bivariate model. A drawback is the need to set the order $m$, which, if set non-optimal, can result in large estimation errors.

Granger causality is then typically defined as the natural logarithm of the ratio of two prediction error variances (Ding et al., 2006), $\sigma_\epsilon^2$ and $\sigma_\eta^2$ for the univariate and bivariate model, respectively:

$$GC_{X \to Y} = \ln \frac{\sigma_\epsilon^2}{\sigma_\eta^2} \tag{3}$$

The null hypothesis of $X$ causing $Y$ (or vice versa), can be tested for significance against a preset p-value, typically 5%. Thus, if $GC_{X \to Y}$ exceeds the preset threshold, assuring that $\sigma_\eta^2$ is significantly smaller than $\sigma_\epsilon^2$, $X$ is said to have a causal effect
on $Y$. Similarly, the causal effect of $Y$ on $X$ can be determined. Note that as the effect of autocorrelation is removed, a simple correlation between $X$ and $Y$ does not guarantee the presence of Granger causality as co-movement does not necessarily imply causality (Aldrich, 1995).

This framework can also be extended to the multivariate case, where the effect of predictors $X$, $Z_1$, $Z_2$ ... $Z_p$ (with $p+1$ the number of predictor variables) on $Y$ can be evaluated. In order to determine the effect of $X$ on $Y$ in a multivariate case, the
performance of a model containing all predictors is compared against that of a multivariate model from which $X$ is excluded, as given by:

$$y_t = \sum_{i=1}^{m} a_i y_{t-i} + \sum_{j=1}^{p} (\sum_{i=1}^{m} b_{j,i} z_{j,t-i}) + \epsilon_{x,t} \tag{4}$$

$$y_t = \sum_{i=1}^{m} a_i y_{t-i} + \sum_{j=1}^{p} (\sum_{i=1}^{m} b_{j,i} z_{j,t-i}) + \sum_{i=1}^{m} c_i x_{t-i} + \eta_t \tag{5}$$

The added value of incorporating $X$ in the set of predictors ($Z_1$, $Z_2$ ... $Z_p$) to improve the prediction of $Y$ can be expressed
in terms of Granger causality as:

$$GC_{X \to Y} = \ln \frac{\sigma_{\epsilon_x}^2}{\sigma_\eta^2} \tag{6}$$

### 2.2.2    Spectral Granger Causality

Despite traditional Granger causality being capable of addressing short-term interactions, simply aggregating time series to their seasonal and annual equivalents prior to following a traditional Granger causality approach does not necessarily lead to

realistic causation inference at larger temporal scales. Consequently, Granger causality frameworks that are defined in the time domain – such as the framework by Papagiannopoulou et al. (2017a), are not designed to capture low-frequency processes. To assess temporal scale-dependent processes, transforming the data into a frequency-dependent domain is crucial as it allows for a differentiation of interactions active at various temporal scales. Therefore, we propose the use of CSGC, which enables to simultaneously condition for other predictors, thus factoring out co-dependency among variables, while addressing processes active at different scales.

The Spectral Granger Causality (SGC) is a non-parametric extension of the Granger causality theory in which time series are first transformed into a frequency domain, resulting in a spectral analogue of Granger causality (Geweke, 1982). A well-known example of such a transformation is the Fourier transformation, where a time series is decomposed in a space solely consisting of frequency. This allows for highlighting strong spectral features, but comes at the cost of time localisation, i.e. the ability to differentiate between processes active at different times. To prevent the loss of the time dimension, SGC adopts a wavelet transformation, which decomposes the original time series into a time–frequency space, thus allowing for both spectral (i.e. temporal scale-dependent) evaluation and time localisation of interactions between predictors and target variable. In order to perform the time–frequency decomposition, the Morlet wavelet is used and a balance between the time and frequency resolutions is obtained by setting the shape parameter to a value of 6, as in Torrence and Compo (1998), or Casagrande et al. (2015). Moreover, to overcome the limitation of assigning an arbitrary order of the system given by Eq. 1 and 2, Dhamala et al. (2008) developed a non-parametric method to express Spectral Granger Causality based on spectral properties of the variables without the need to estimate the model order, given by:

$$SGC_{X \to Y}(f) = \ln \frac{S_{yy}(f)}{S_{yy}(f) - \left[\Gamma_{xx} - \left(\frac{\Gamma_{yx}^2}{\Gamma_{yy}}\right)\right]|H_{yx}(f)|^2} \tag{7}$$

where $S_{yy}(f)$ equals the spectral density (power spectrum) of the target variable $Y$ at frequency $f$, which can be estimated from the wavelet transform. Using the variables $X$ and $Y$, the error covariance matrix $\Gamma$ and the spectral transfer function matrix $H(f)$ can be calculated using matrix factorisation (Wilson, 1972). For more information on SGC, we refer to Ding et al. (2006), Dhamala et al. (2008), and Detto et al. (2012, 2013).

### 2.2.3 Conditional Spectral Granger Causality

Eq. 7 is only valid to determine the effect of a variable $X$ on $Y$, without taking into account that other variables might influence both the predictor and target, consequently inducing an apparent causal relationship. To tackle this issue, conditionality between variables has to be taken into account, for which the SGC framework can be extended to Conditional Spectral Granger Causality (CSGC). In other words, SGC can be adapted to CSGC to assess if $X$ causes $Y$ given that $Z_1$, $Z_2$ ... $Z_p$ may cause $Y$ and $X$, resulting in a conditioned measure of spectral causality $CSGC_{X \to Y|Z_1,Z_2...Z_p}(f)$. For a multivariate problem with p+2 variables ($Y$, $X$, $Z_1$, $Z_2$ ... $Z_p$), the system can be written, after spectral transformation and Wilson factorisation (Wilson,

1972), as:

$$\mathbf{S}(Y, X, Z_1, Z_2...Z_p, f) = \mathbf{H}(f)\,\mathbf{\Sigma}\,\mathbf{H}^*(f) \tag{8}$$

$$\mathbf{U}(Y, Z_1, Z_2...Z_p, f) = \mathbf{G}(f)\,\mathbf{\Gamma}\,\mathbf{G}^*(f) \tag{9}$$

with $\mathbf{S}$ and $\mathbf{U}$ representing the spectral matrices of the complete system and the system with the variable whose causality is tested being excluded, i.e. $X$ in this case, respectively. Similarly, $\mathbf{H}$ and $\mathbf{G}$ are the spectral transfer function matrices, while $\mathbf{\Sigma}$ and $\mathbf{\Gamma}$ equal the error covariance matrix of the full and incomplete system of variables, respectively, and where $^*$ indicates matrix adjoint.

From Eq. 8 and 9, CSGC of $X$ on $Y$ given $Z_1, Z_2 ... Z_p$ can be calculated as:

$$CSGC_{X \rightarrow Y|Z_1, Z_2...Z_p}(f) = \ln \frac{\Gamma_{yy}}{|Q_{yy}(f)\Sigma_{xx}Q_{yy}^* f|} \tag{10}$$

where:

$$\mathbf{Q} = \begin{pmatrix} \tilde{G}_{YY} & 0 & \tilde{G}_{YZ_1} & ... & \tilde{G}_{YZ_p} \\ 0 & 1 & 0 & ... & 0 \\ \tilde{G}_{Z_1Y} & 0 & \tilde{G}_{Z_1Z_1} & ... & \tilde{G}_{Z_1Z_p} \\ ... & ... & ... & ... & ... \\ \tilde{G}_{Z_pY} & 0 & \tilde{G}_{Z_pZ_1} & ... & \tilde{G}_{Z_pZ_p} \end{pmatrix}^{-1} \begin{pmatrix} \tilde{H}_{YY} & \tilde{H}_{YX} & ... & ... & \tilde{H}_{YZ_p} \\ \tilde{H}_{XY} & ... & ... & ... & ... \\ ... & ... & ... & ... & ... \\ ... & ... & ... & ... & ... \\ \tilde{H}_{Z_pY} & \tilde{H}_{Z_pX} & ... & ... & \tilde{H}_{Z_pZ_p} \end{pmatrix} \tag{11}$$

In Eq. 11, $\tilde{\mathbf{H}}(f) = \mathbf{H}(f)\mathbf{P}^{-1}$ and $\tilde{\mathbf{G}} = \mathbf{G}\mathbf{P}_2^{-1}$ represent corrected transfer function matrices to separate the directional interactions (Geweke, 1982). The rotation matrices $\mathbf{P}$ are normalisation matrices needed to transform the multivariate systems in their canonical form with uncorrelated errors (Detto et al., 2013). For more information on CSGC, we refer to Dhamala et al. (2008) and Detto et al. (2012, 2013).

Using Eq. 10, Conditional Spectral Granger Causality of $X$ on $Y$ can be determined, given the influence of $Z_1, Z_2 ... Z_p$ on both $X$ and $Y$. If $X$ is not directly affecting $Y$, but for example $Z_1$ is forcing both $X$ and $Y$, the numerator in Eq. 10 will equal the denominator, thus resulting in a Granger causality measure of zero. However, if there is a direct causal influence of $X$ on $Y$ at a specific frequency $f$, $CSGC_{X \rightarrow Y|Z_1, Z_2...Z_p}(f) > 0$. Using Eq. 10, it is possible to determine if $X$ (Granger-) causes $Y$, but no information on the sign of the causal relation can be extracted.

### 2.2.4 Significance testing of CSGC

Despite the ability of Eq. 10 to account for conditional effects between variables, it fails to determine how robust the found interactions are. Therefore, the robustness of the determined CSGC values needs to be tested against the null hypothesis that $X$ has no causal effects on $Y$. In case of Granger causality in the time domain, significance of the determined statistic, e.g. $GC$, can be tested by a bootstrapping scheme in which the time series are randomly shuffled before determining the $GC$-values. By repeating this procedure $n$ times, the distribution of $GC$ can be determined. By selecting a p-value, typically 5%, the determined Granger causality of $X$ on $Y$ can be tested against the null hypothesis of no causal interaction.

However, for the spectral variant of Granger causality, a simple randomisation of the time series induces unwanted artefacts. Due to the spectral nature of the method, the power spectrum of the randomised time series must be preserved, i.e. to be equal to that of the original time series at each frequency. In other words, if the original time series are characterised by much high-frequency variation and less at lower frequencies, the time series used for significance testing need to show the same frequency-dependent variability. Therefore, surrogate time series exhibiting the same spectral power as the original time series need to be used. Here, iterative amplitude adjusted Fourier transform (IAAFT) surrogates are used in combination with Monte Carlo simulations, as CSGC is non-parametric (Detto et al., 2012), to test the determined CSGC-value against the null hypothesis of no causal interaction. Due to computational constraints, 100 runs with surrogates were performed for each set of original time series (i.e. for each pixel), and will be used to test for significance (p-value < 0.05). However, to increase the robustness of the results, an ensemble of products is used for both the observations and models as explained in Sect. 2.1.

### 2.2.5   Explained variance

CSGC, as defined by Eq. 10, compares the performance of two autoregressive models in explaining variation in a target variable $Y$. In other words, does $X$, given a set of predictors $Z_1$, $Z_2$ ... $Z_p$, improve the estimate of $Y$ compared to a model that only uses $Z_1$, $Z_2$ ... $Z_p$. In this study, we are interested in quantifying how much of variance in the target variable is actually directly explained by a predictor, and not how much did the estimation error improve upon adding $X$ to the set of the predictors. Therefore, we deviate from the traditional formulation of Granger causality and define a new measure, the fraction ($F$) of variance in the target variable $Y$ that is explained by a predictor $X$. Ideally, the new formulation would be:

$$F_{X \to Y} = \frac{\sigma_X^2}{\sigma_Y^2} \times 100\% \tag{12}$$

with, $\sigma_Y^2$ representing the total variance of $Y$ and $\sigma_X^2$ the variance in $Y$ explained by $X$. However, a part of the variance in $Y$ is not explainable by any predictor, as is forced by the autocorrelation of $Y$ ($\sigma_{Y,auto}^2$). Therefore, in order to account for the part of variance in $Y$ that will not be able to be explained by any predictor, Eq. 12 is adapted to:

$$F_{X \to Y} = \frac{\sigma_X^2}{\sigma_Y^2 - \sigma_{Y,auto}^2} \times 100\% \tag{13}$$

As traditional Granger causality and CSGC determine a measure of causality that is defined in a similar way, Eq. 1 can be used to determine how $F$ can be calculated from the actual Granger causality value. Considering the univariate model given by Eq. 1, the total variance in the target variable $Y$ can be rewritten as:

$$\sigma_Y^2 = \sigma_{Y,auto}^2 + \sigma_\epsilon^2 \tag{14}$$

with $\sigma_\epsilon^2$ representing the unexplained variance or prediction error variance. Substituting Eq. 14 into Eq. 13 results in:

$$F_{X \to Y} = \frac{\sigma_X^2}{\sigma_\epsilon^2} \times 100\% \tag{15}$$

This derivation can also be extended towards the multivariate case, and even to CSGC. As Eq. 15 equals $1 - e^{-GC_{X \to Y}}$, the conditional spectral variant of the fraction of variance in $Y$ explained by $X$ can be calculated as:

$$F_{X \to Y | Z_1, Z_2 \ldots Z_p}(f) = \frac{\Gamma_{yy} - |Q_{yy}(f) \Sigma_{xx} Q_{yy}^* f|}{\Gamma_{yy}} \times 100\% \tag{16}$$

Using Eq. 16, the impact of climate on vegetation and the feedbacks of vegetation on climate can be quantified and reported in
an intuitive manner (see Fig. 1a and Sect. 3).

### 2.2.6 Determining scales of interest

As pointed up in Sect. 1, monthly interactions between climate and vegetation have been studied by many authors (Nemani et al., 2003; Wu et al., 2015; Papagiannopoulou et al., 2017b). On the other hand, the phenological cycle or inter-annual variability of climate and vegetation are also expected to interact, yet little is known about how these interactions differ from the
short-term processes. Hereafter, the terms *phenology* and *phenological cycle* are used to refer to the seasonal-scale variability in LAI. This reflects features such as the timing of the growing season or the amplitude of the intra-annual cycle (Richardson et al., 2009; Verger et al., 2016), since CSGC will react to variability in both the time and frequency domains. As explained in Sect. 2.2.3, CSGC allows a simultaneous analysis of the interactions at multi-temporal scales, while no assumption needs to be made about the direction of the interplay between climate and vegetation. Moreover, based on the characteristics of the climate
data used in this study, CSGC can be applied to assess causality over a wide range of temporal scales, starting at 2 months (twice the temporal resolution) and going up to 16.5 years (maximum temporal scale due to discretisation of the frequency space; can be adjusted if needed, especially for longer time series).

In order to determine which range of temporal scales better represents monthly, seasonal and inter-annual interactions, an experiment with synthetic monthly time series was performed. First, a predictor variable ($X1$) is constructed with imposed
variability at the scales of interest (e.g. monthly, seasonal and inter-annual). Monthly variability is assumed to be random from month to month, while seasonality is defined as consecutive three-block periods of constant value. Inter-annual variation is defined as blocks of one year with a fixed value. The predictor $X1$ is constructed by randomly generating these three variabilities and adding them. Finally, a linear trend is added to $X1$ to be able to retrieve the maximum scale at which inter-annual variability can be observed. Next, a target variable ($Y1$) is constructed with a known causal relation to the predictor
$X1$ by multiplying $X1$ with a random factor and then shifting $Y1$ in time so that $Y1$ lags $X1$ by one month. Using these two synthetic time series, SGC is used to determine the Granger causality of $X1$ on $Y1$. Note that SGC is used instead of CSGC as the scales at which the targeted interactions can be observed are identical for a bivariate and multivariate case.

In order to identify the scales that are most sensitive to monthly, seasonal and inter-annual interactions, a new predictor variable $X2$ is constructed as an identical copy of $X1$, except for one specific variability. For example, if the range of scales
that capture monthly interactions is determined, $X2$ will be equal to $X1$, but with perturbed monthly variability. Next, a new target variable $Y2$ is constructed by multiplying $X2$ with a new random factor and again guaranteeing that $Y2$ lags $X2$ by one month. Then, SGC is used to determine if $X1$ Granger causes $Y2$, which will show a decrease in Granger causality at scales that capture the perturbed interaction compared to the Granger causality of $X1$ on $Y1$. Consequently, by repeating

this procedure for all the interactions that are to be assessed (i.e. monthly, seasonal and inter-annual), comparison of the two Granger causalities allows to record the range of scales that capture these interactions. To increase robustness, this procedure is repeated 100,000 times, resulting in a clear delineation of scales representing monthly (0–0.32 years), seasonal (0.32–1.54 years) and inter-annual (1.54–9 years) interactions. Decadal patterns of trends cannot be investigated here due to length of

the observational record (see Sect. 2.1), but are used in the determination of the ranges to fix the upper limit for inter-annual interactions. See Fig. 1b for an illustration of the resulting scales, which are considered to be time- and space-invariant. Results will be presented as mean patterns for each scale using the determined ranges. Selecting the maximum explained variance within each range, unwillingly results in the taking the CSGC at the highest scale of each interval, as the CSGC increases with the scale (for more information, see Sect. 3.1).

# 3 Results and discussion

## 3.1 Climate impact on vegetation in observations

Fig. 2a,c,e illustrate the Granger causality of air temperature, net radiation and precipitation on LAI dynamics, based on observations, globally and latitudinally. Results are shown separately for monthly (Fig. 2a), seasonal (Fig. 2c) and inter-annual (Fig. 2e) time scales using a tri-variate colormap according to the fraction explained by each climatic driver (see Sect. 2.2.5).

Dotted pixels indicate that in at least 75% of the ensemble members there is (a) agreement regarding the dominant climate impact, and (b) statistical significance (at the 5% level). At monthly scales, overall spatial patterns in the observation-based results (Fig. 2a) are in agreement with previous studies, showing the dominance of precipitation in arid and semi-arid regions, while radiation and temperature dominate in northern latitudes and rainforests, respectively (Nemani et al., 2003; de Jong et al., 2013; Seddon et al., 2016; Papagiannopoulou et al., 2017b). Strong radiation effects on vegetation can be observed

over northern latitudes due to severe limitations in incoming radiation during winter months. However, in those latitudes, LAI retrievals are contaminated by snow cover signals. While focusing on the growing season could solve this issue, the CSGC requires continuous time series. Because in wintertime, due to limitations in solar radiation, plant growth is inhibited in northern latitudes, most variability captured at monthly scales will be dominated by the more dynamic spring and summer periods; therefore, our results suggests that radiation still dominates the behaviour of vegetation at these latitudes.

This dominant high-latitude radiation control was not reported by Papagiannopoulou et al. (2017b), who, based on a non-linear Granger causality framework, found that 61% of the vegetated land surface is primarily driven by water availability at monthly time scales, while temperature and radiation are the primary factors in only 23% and 15% of the vegetated surface, respectively. These results also contrasted with earlier studies, that pointed to a less dominant role of water availability for global ecosystems (Nemani et al., 2003; Wu et al., 2015). Here, our monthly-scale results also show a dominant role of

precipitation, yet more moderate; 51% of vegetated land is primarily controlled by precipitation, with radiation being the primary control factor in 40% as well. When the analysis targets vegetation anomalies by detrending linearly and subtracting the average seasonal cycle for both LAI and climate (as was done in Papagiannopoulou et al. (2017a, b)), results show a similar dominance of precipitation, but air temperature gains importance over net radiation (being the dominant driver over 13%, and

36%, respectively, as indicated in Supplementary Fig. A1). The higher importance of water availability in Papagiannopoulou et al. (2017b) can be attributed to accounting directly for the effect of (root-depth) soil moisture as a driver of vegetation, as opposed to the use of precipitation only in this study. Also, human practices, such as i.e. irrigation, can potentially bias our results. Nonetheless, irrigation is expected to increase the energy-dependence of LAI dynamics, and as irrigation tends to be a seasonal phenomenon restricted to the growing period, this increase is found to be clearer at seasonal than monthly scales (as shown in Supplementary Fig. B1). A final difference with Papagiannopoulou et al. (2017b) is the consideration in the latter of snow water equivalent as a water availability driver, which explains the divergence with our results in higher latitudes. Our results can also be reconciled with previous studies, such as Nemani et al. (2003), Wu et al. (2015), and Seddon et al. (2016); regional differences may relate to the specific focus of those studies on one temporal scale only, their calculation of covariances instead of inferring causality in a more formal manner, or the use of different variables to assess water availability drivers.

As mentioned before, a key feature of CSGC is that it also enables the assessment of interactions at longer temporal scales, such as seasonally (Fig. 2c) and inter-annually (Fig. 2e). As expected, radiation is found to dominate the seasonal phenology over 55% of the global vegetated land. The strong radiation control over northern latitudes is attributed to the amplitude of the solar cycle, which ultimately inhibits vegetation growth during wintertime. In this analysis, net radiation instead of incoming radiation has been used, in order to be consistent with the investigation of vegetation–climate feedbacks in Sect. 3.3; however, using incoming radiation as driver instead, leads to a similar 54% dominance (see Fig. C1). Compared to monthly scales, seasonal precipitation control is less widespread, as only 33% of the vegetated land is primarily controlled by precipitation (compared to 51% at monthly scales; Fig. 2a and 2c). This reduced importance of precipitation can be attributed to the observed temperature-driven hotspot in the Sahel region, but more importantly to increase of radiation control over the south of Eurasia and in tropical forests. Furthermore, the patterns in Amazonia tend to agree with the findings of Saleska et al. (2007, 2016), Phillips et al. (2009), and Hilker et al. (2014), showing a dominance of water availability in the southeastern side, while radiation is more limiting in the northwest.

Finally, at inter-annual scales, despite co-dominance of multiple drivers in some regions, global ecosystems tend to be water limited with 43% of the vegetated land surface being primarily dominated by precipitation (Fig. 2e), especially in the subtropics. Although patterns exhibit some heterogeneity, not only arid and semi-arid regions show a (significant) dominant control by precipitation, but also substantial parts of southern Eurasia. This widespread inter-annual dependency on water availability of ecosystem dynamics may arise due to the large inter-annual variability of precipitation, and has already been documented in relation to the impact of precipitation of global carbon budgets (Poulter et al., 2014) and terrestrial evaporation (Miralles et al., 2014). Moreover, it agrees with the results of Green et al. (2019) and Humphrey et al. (2018), yet it does not necessarily contradict the findings by Jung et al. (2017); the latter reported a dominant role of temperature at the global scale, yet showed a dominance of water availability at regional scales that is compensated when upscaling to global means. Inter-annually, the control of air temperature extends over the high northern latitudes and eastern China, dominating in 20% of vegetated land, while radiation remains the most crucial driver for 37% of the land surface, almost exclusively in the northern latitudes, likely affected by the strong seasonal patterns (Fig. 2e). Once the seasonality is removed, the inter-annual dominance of radiation control falls down to 20% of the vegetated land surface (see Fig. A1c). Despite the heterogeneity, the overall control of climate

on vegetation is higher at inter-annual scales than at shorter time scales, as can be observed in the latitudinal profiles, which show the total causality in absolute terms (Fig. 2). This is partly a consequence of the time–frequency decomposition of CSGC, which generally results in higher values of explained variance at longer time scales due to the increased time frame over which a predictor variable is assessed, thus increasing the chance of incorporating memory effects. However, the significance test against the null hypothesis of exhibiting no causal effect, ensures that regions exhibiting significant responses can be compared over different time scales.

Noteworthy is that anthropogenic effects, which are not directly addressed here, can also impact vegetation and climate at short temporal scales. For example, irrigation and deforestation can result in a decoupling between climate and vegetation (Lawrence and Vandecar, 2015; Chen et al., 2019). In the tropics, deforestation results in a warming effect due to reduced plant transpiration, which in turn may induce a decline in precipitation, creating a warmer and drier regime (Lawrence and Vandecar, 2015). Irrigation allows for growing crops in water-limited regions, consequently inducing energy constraints which are captured by the CSGC. Note that due to the limited data record, the effects of global warming trends and carbon dioxide fertilisation – and the consequent trends in vegetation greening and water use efficiency (Reichstein et al., 2013; Wu et al., 2015; Zhu et al., 2016) – are not directly addressed in this study.

## 3.2 Climate impact on vegetation in models

Results of the observations are next used to benchmark CMIP5 ESM performance in representing the control of climate on vegetation (Fig. 2b, 2d, 2f). Dotted pixels indicate that at least three out of four models reach agreement regarding (a) dominant climate impact, and (b) statistical significance (at the 5% level). Comparison of Fig. 2a and 2b shows that the monthly impact of air temperature on ecosystems is strongly overestimated by ESMs, with 17% and 26% of vegetated land being primarily dominated by temperature for observations and ESMs, respectively. This coincides with a lower effect of net radiation in central Eurasia and, more importantly, elevated air temperature control in the Amazon and Congo rainforests. These contrasting results with observations might hint towards problems in ESMs with respect to representing the behaviour of the tropics, but may also relate to the difficulties to retrieve LAI from satellites in dense forests (Hilker et al., 2015). Nevertheless, ESMs agree on the general patterns that highlight the strong radiation effects in northern latitudes (albeit less extended), and the water availability as main driver in arid and semi-arid regions at monthly time scales.

Seasonally, a larger control of precipitation and air temperature on vegetation phenology is also noticeable over the equator for ESMs (see latitudinal profile in Fig. 2d). The dominant control of radiation on vegetation phenology over northern latitudes is similar for all models (inter-model agreement and significance represented by the black dotting), and whereas the spatial extent agrees with the observational results, the magnitude is underestimated by the models (see Fig. 2c and 2d). Radiation is the primary driver of the seasonal LAI variation in 45% of the vegetated land in models (compared to 55% for the observations). The role of precipitation and air temperature as drivers of the phenological cycle gains in importance in ESMs, at the cost of radiation, with 40% and 15% of seasonal LAI variation being dominated by precipitation and air temperature variability, respectively, versus the 33% and 12% in observations, respectively. Despite the overall similarities in the patterns of dominant drivers, regional differences between observations and models are still observed. Models point towards a water-limited phe-

nological cycle in the Sahel, while observations hint also towards a dominant role of temperature (compare Fig. 2c and 2d). Furthermore, whereas observations clearly highlight a south-to-north water-to-energy-limited gradient in Amazonia, models tend to disagree and point towards temperature as a key driver over most of the Amazonian rainforest at seasonal scales. These differences might indicate difficulties to model climate–vegetation interactions across the basin where air temperature is found

to be the only limiting control, yet they may again be influenced by the difficulties to retrieve LAI from satellites over dense canopies, as pointed above.

Similar to observations, the climate impact on LAI increases with longer temporal scales in ESMs. However, more remarkable than in the observations is the strong water limitation across the globe at inter-annual scales, which is not restricted to arid and semi-arid regions (Fig. 2f). Water availability at inter-annual scales is dominant for vegetation over 62% of land versus

the 43% found in observations (Fig. 2e), and is also strongly overestimated in absolute terms at most latitudes, especially in the tropics. Further analysis shows that the divergence in the considered period between observations and models (see Sect. 2.1) does not substantially impact results; repeating the analysis for the overlapping time range for observations and models (1982–2005) yields very similar findings (Fig. D1).

### 3.3   Vegetation feedback on climate in observations and models

Analogous to the effect of climate on vegetation, vegetation can alter local (and remote) climate conditions via biophysical and biochemical feedbacks. These feedbacks arise from the effect of vegetation structure and physiological activity on the surface radiation budget, available energy partitioning into latent and sensible heat fluxes, aerodynamic conductance of the ecosystem, atmospheric chemical composition, and indirect processes affecting incoming radiation, atmospheric humidity and temperature (McPherson, 2007; Bonan, 2008). The representation of these feedbacks in ESMs remains in need for improvement

to accurately predict future climate (de Noblet-Ducoudré et al., 2012; Zhang et al., 2016). Here, we unravel these feedbacks of LAI on different climate variables based on observations (Fig. 3a, 3c, and 3e) and ESM data (Fig. 3b, 3d, and 3f), and at different temporal scales, from monthly (Fig. 3a and 3b), to seasonal (Fig. 3c and 3d) and inter-annual (Fig. 3e and 3f). Dotted pixels indicate that in at least 75% of the ensemble members there is (a) agreement regarding the dominant feedback, and (b) statistical significance (at the 5% level). To aid comparison to the strength of climate impacts on vegetation – measured in

relative or absolute percentage of caused variance (see Sect. 2.2.5) – an identical tri-variate colormap to that in Fig. 2 is used.

Observed LAI feedbacks over the mid and high northern latitudes concentrate on surface net radiation at monthly time scales (Fig. 3a). As vegetation lowers the albedo in boreal regions, it allows for more energy storage and less reflection back into the atmosphere; this increases surface net radiation and may lead to a net warming effect (e.g. Bonan, 2008; Forzieri et al., 2017). By repeating the analysis using only incoming (shortwave and longwave) radiation, instead of surface net radiation, the results

indicate that the influence of LAI on (e.g.) cloud formation is limited, at least considering the local (in the sense of 'spatially collocated') scales revealed by the causal framework (see Fig. E1). Monthly feedbacks of vegetation on precipitation and air temperature are spatially less widespread, however, significant feedbacks on precipitation are observed, especially in tropical forests. The patterns in Amazonia suggest a more dominant effect of vegetation on radiation in the north, while precipitation feedbacks dominate in the south (Fig. 3a). We note that the method does not differentiate whether higher or lower values of LAI

cause more or less rainfall, only that a causal effect of LAI on rainfall exists. The south-to-north patterns in the Amazon agree with the larger dependency on precipitation recycling in the South (Dirmeyer et al., 2009; Zemp et al., 2014). Tropical forests are known to regulate local (and global) precipitation as their large use of water increases atmospheric humidity and results in cloud formation (Malhi et al., 2008). This also directly affects the incoming short- and long-wave radiation. Nevertheless,

we restate that the method only focuses on the effects of LAI on its immediate climatic environment, not in neighbouring or remote locations.

At seasonal scales, an increase of feedbacks on temperature is observed in the Northern Hemisphere, and feedbacks on precipitation remain limited to the tropics, although practically no statistical significance is reached outside the tropics (Fig. 3c). Finally, at inter-annual scales, the observation-based results show a north-to-south gradient over the Sahel region, with the

north exhibiting feedbacks on precipitation, while strong vegetation feedbacks on temperature are observed in the south (Fig. 3e). However, despite the highly significant interactions in the tropics, and except for the feedback on radiation in the Northern Hemisphere, the inter-annual feedbacks cannot be clearly disentangled using the CSGC, as shown by the incoherent spatial patterns in Fig. 3e. This may occur due to the long integration time and the somehow limited observational record. Individual ensemble members do achieve high significance, but little inter-product agreement is reached due to high spatial heterogeneity

over ensemble members. Overall, and as expected, comparisons between Fig. 2 and 3 reveal that the impact of climate on vegetation consistently exceeds the strength of the vegetation feedback on climate. This means that local climate variability leaves a larger imprint on LAI dynamics than vice versa. This can be partly attributed to the fact that only local interactions are considered here: while vegetation reacts to its most immediate environment, vegetation can lead to remote effects on climate that are not addressed in our analyses (Dirmeyer et al., 2009; Miralles et al., 2019). Nevertheless, these results show

the importance of LAI variability in explaining the variance in local climate at intra-annual scales – mainly through impacts on the net radiation induced by albedo changes – and the potential of the CSGC framework to disentangle the bi-directional interaction between vegetation and climate.

In general, ESMs seem to correctly capture the spatial extent of LAI effects on net radiation throughout most of the Northern Hemisphere, but underestimate feedbacks of vegetation on air temperature, which originates from either an actual underesti-

mation of the air temperature feedback by ESMs, or an overestimation of the feedback on net radiation in these regions, as reported by Forzieri et al. (2018) and confirmed by the latitudinal profiles (Fig. 3b,d,f), which masks the vegetation feedback on air temperature. Despite the overestimation, models do agree with each other on the influence of LAI on net radiation at polar latitudes (see dotted pixels), and the overall mean ensemble patterns for monthly and seasonal time scales also agree with observational results. Interestingly, while observations show significant impacts of LAI on precipitation in the (sub-)tropics,

these effects are not entirely reproduced by ESMs, which tend to show a larger influence of LAI on temperature in those regions. This may suggest a lower dependency of tropical forests on rainfall recycling (Malhi et al., 2008; Hilker et al., 2014; Zemp et al., 2017) and/or an overall wet bias in the ESMs (Mueller and Seneviratne, 2014); the latter is however not supported by the results in Fig. 2 that indicate an overall overestimation of water limitations in models. Nonetheless, these local feedbacks on temperature and precipitation are overall weak – both in observations and models – as indicated by the absolute magnitudes

shown in the latitudinal profiles (Fig. 3).

## 3.4 Biome-specific interactions

Finally, to better visualise the multi-temporal scale vegetation–climate interactions in observations and models, results are presented averaged per biome type. Fig. 4 shows the biome-averaged absolute observed and modelled climate control on LAI dynamics, while Fig. 5 presents the vegetation feedbacks on climate. Forest ecosystems are generally found to be energy-
driven, in agreement with previous studies (Nemani et al., 2003; Seddon et al., 2016; Papagiannopoulou et al., 2017b). ESMs tend to agree with the observations on the magnitude of the response of ecosystems to radiation at all temporal scales, with the exception of the over-sensitivity of evergreen broadleaf forests (EBF) at monthly scales and for most models. In regards to the influence of air temperature, strong differences with observations can be noticed at seasonal time scales for forest biomes; this is most remarkable for broadleaf forests, both evergreen and deciduous (EBF and DBF), which show a model overestimation
of the control of temperature on LAI dynamics, even for the minimum modelled temperature control. Interestingly, models also overestimate the sensitivity of broadleaf forests (EBF and DBF) to precipitation, especially at inter-annual time scales. Observation results show limited water stress in tropical and mid-latitude forests, arguably due to the deep rooting system and mild climate. However, this apparent model over-dependency of broadleaf forests on climate may also emerge from the under-sensitivity of the observational results due to the saturation of the greenness signal received by satellites in dense canopies.
Models unambiguously overestimate the importance of water availability for LAI in most biome types at inter-annual time scales, and to a more limited extent at monthly and seasonal scales – this appears in contrast with the results of Green et al. (2017). As expected, savannas are found to be mainly driven by precipitation across all time scales both in observations and models, although models strongly disagree among each other, as reflected by the large error bars in Fig. 4.

On the other hand, short-term feedbacks of LAI on climate seem to be better represented in ESMs, as small differences can be
seen when compared to the observational results in Fig. 5. Note that this statement only holds true if looked at biome-averaged patterns due to compensatory effects, as comparison of observations and models in Fig. 3 does indicate to clear regional differences. Deciduous needleleaf forests (DNF) and evergreen needleleaf forests (ENF) exhibit the strongest feedback on net radiation (and temperature) at all temporal scales; once again this appears related to albedo changes and not impacts on, e.g., cloud formation (see Fig. E1). Nonetheless, the effect of needleleaf forests on the radiation budget tends to be overestimated by
most CMIP5 models, especially at monthly and seasonal time scales, which aligns with the findings of Forzieri et al. (2018). ESMs also overestimate the influence of ecosystem phenology on net radiation in mixed forests (MF), open shrublands (OS), and woody savannas (WS); yet, large inter-model disagreements exist on the seasonal influence of LAI on net radiation for almost all biomes, as illustrated by the large error bars Fig. 5. The strength of the effect of LAI on precipitation is overall lower than its impact on net radiation and air temperature, partly due to the non-consideration of downwind influences in this analysis
which have been shown to be crucial (Dirmeyer et al., 2009; Zemp et al., 2017). However, similar to the results of Green et al. (2017), a strong influence of LAI on precipitation can be observed in savannah regimes.

## 4 Conclusion

Here, bi-directional interactions between climate and vegetation in global remotely-sensed observations were analysed at different temporal scales using Conditional Spectral Granger Causality (CSGC) with the aim to benchmark the representation of these interactions in ESMs. Three main climate variables are considered, namely air temperature, net radiation and precipitation, while LAI is used as a proxy for vegetation state. While CSGC is not in principle designed to cope with non-linear interactions, it has the advantage of being able to assess both the climate impact on vegetation and the vegetation feedback on climate, while differentiating simultaneously between different temporal scales. Our findings for monthly interactions agree with those of earlier studies (Nemani et al., 2003; Wu et al., 2015; Papagiannopoulou et al., 2017b), with (semi-)arid regions showing a primary control by water-availability, while the tropics and high northern latitudes being primarily energy-limited. Fig. 6 gives an overview of the overall global interactions between climate and biosphere. Averaged over all vegetated land, radiation is found to dominate vegetation dynamics at seasonal scale, but models seem incapable of reproducing the observed spread in the strength of this dependency. ESMs generally overestimate the precipitation control on vegetation, and most drastically at inter-annual scales. On the other hand, vegetation feedbacks are found to be locally more predominant for net radiation over all time scales, mainly due to the strong interplay between radiation and vegetation at northern latitudes. As shown by the summary in Fig. 6, ESMs tend to overestimate the feedbacks on the radiation budget, while feedbacks on local precipitation are often underestimated, especially at seasonal and inter-annual scales. Finally, interactions in both ways are found to increase with increasing time scales, and feedbacks of vegetation on climate explain a lower fraction of the variance in the latter than vice versa.

Despite the clear advantages over traditional statistical analysis, the application of CSGC is subject to a series of assumptions. Firstly, CSGC can condition for other variables to exclude effects due to co-dependency, but this implies that the variable has to be considered. Here, we limited the potential drivers of vegetation to air temperature, net radiation and precipitation, but vegetation is also affected by other factors such as nutrient availability, atmospheric carbon dioxide concentrations etc. Second, only local interactions are considered, meaning that interactions are assumed to be spatially collocated. This assumption might be valid for the impact of climate on vegetation, but is certainly an oversimplification regarding the vegetation feedbacks on climate which are rarely of local nature, especially when it refers to cloudiness and rainfall. Finally, despite the use of observation ensembles, errors due to difficulties in retrieving LAI over dense canopies, and biases in LAI products outside the growing season might affect our results. Adapting the causal framework to resolve changes in sensitivities over time would allow the consideration of these and other aspects, and increase the potential of the method to address scientific challenges related to changes in sensitivity of different climate factors over time. That would enable, for instance, a benchmarking of the ESM skill to reproduce changes in ecosystem resilience to climate.

*Code availability.* Our scripts can be accessed via https://github.com/lhwm.

**Appendix A:  Climate impact on vegetation in anomalies of observations**

**Appendix B:  Climate impacts on vegetation as a function of irrigation for observations**

**Appendix C:  Climate impact on vegetation in observations using incoming radiation instead of net radiation**

**Appendix D:  Climate impact on vegetation in observations and ESMs during 1982–2005**

5  **Appendix E:  Vegetation feedback on climate in observations using incoming radiation instead of net radiation**

*Author contributions.* DGM and JC conceived the study and led the writing. JC conducted the analysis. MDem contributed to the data-processing. AM and MDet contributed to the implementation of the method. All co-authors contributed to the design of the experiments, interpretation of results and editing of the manuscript.

*Competing interests.* The authors declare that they have no conflict of interest.

5  *Acknowledgements.* This work is funded by the Belgian Science Policy Office (BELSPO) in the framework of the STEREO III programme, projects SAT-EX (SR/00/306) and SAT-EX Wave (SR/02/367). Diego G. Miralles acknowledges funding from the European Research Council (ERC) under grant agreement 715254 (DRY–2–DRY). We acknowledge the World Climate Research Programme's Working Group on Coupled Modelling, which is responsible for CMIP. We also thank the climate modelling groups for their effort in producing and making available of their model output.

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

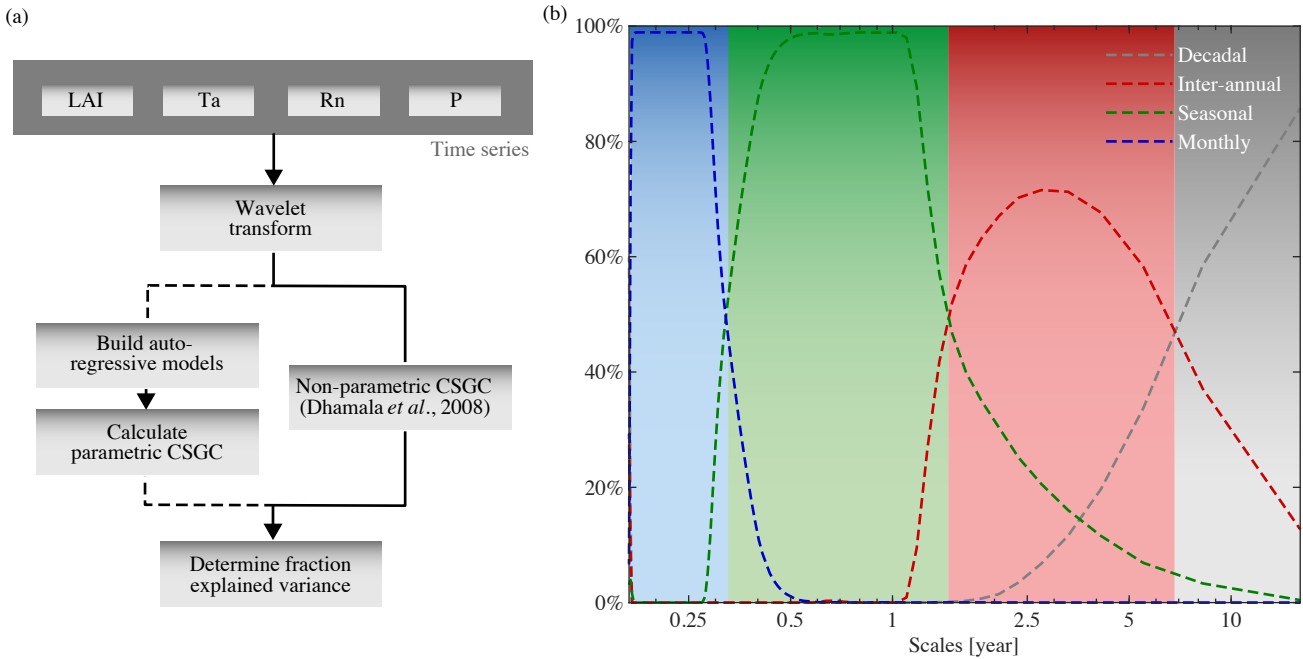

**Figure 1.** **(a)** Schematic overview of CSGC, extended by the calculation of the fraction of explained variance.**(b)** Scales affected by perturbation of variability in synthetic time series at a particular temporal scale. Coloured lines show for each perturbed variability the scales that changed most compared to the unperturbed runs as a percentage of runs out of 100,000. The shaded colours indicate the ranges adopted for each temporal scale in the analysis.

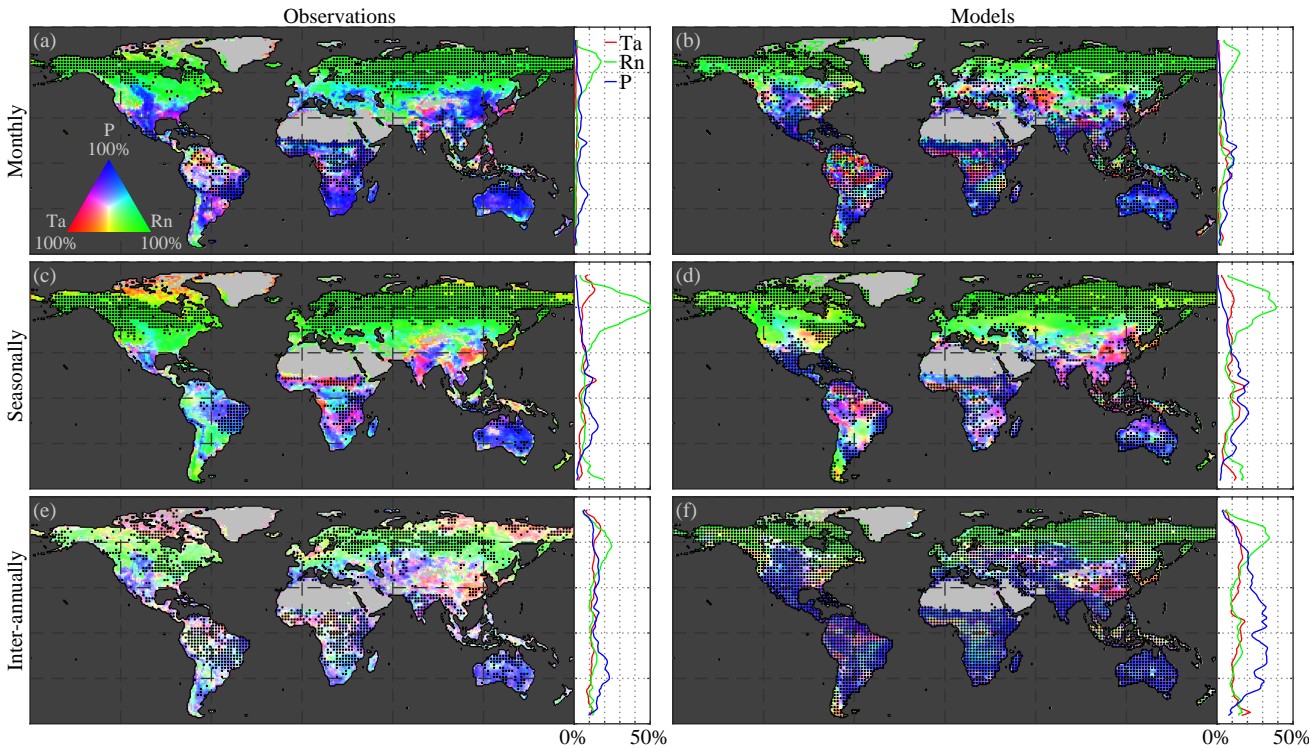

**Figure 2.** Global climate impact on vegetation. Variability in **(a, c, e)** observed and **(b, d, f)** modelled LAI caused by air temperature (Ta), net radiation (Rn) and precipitation (P) at **(a, b)** monthly, **(c, d)** seasonal, and **(e, f)** inter-annual time scales. Maps show the causality in relative terms with respect to the dominant driver at each pixel, while the latitudinal profiles show the absolute impact of each driver. The period 1982–2015 is taken as reference for the observations, while models span 1956–2005. Maps show the mean from the ensemble of the observations or four CMIP5 models: CCSM4, HadGEM2-ES, NorESM1-M, IPSL-CM5A-MR. Dotted pixels indicate a significant (p-value = 5%) primary driver agreed upon by at least 75% of the ensemble members.

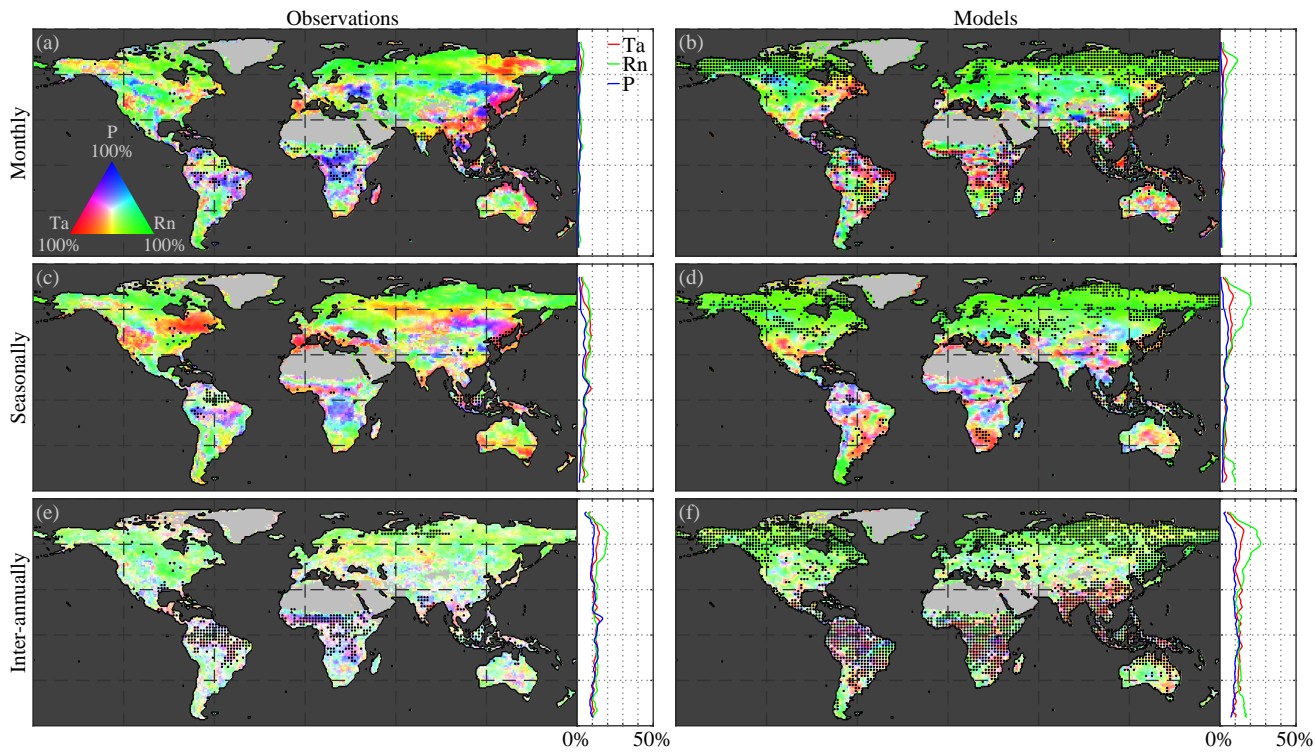

**Figure 3.** Global vegetation feedback on climate. Variability in air temperature (Ta), net radiation (Rn) and precipitation (P) that is caused by ((**a, c, e**) observed and (**b, d, f**) modelled LAI at (**a, b**) monthly, (**c, d**) seasonal, and (**e, f**) inter-annual time scales. Maps show the causality in relative terms with respect to the strongest feedback at each pixel, while the latitudinal profiles show the absolute feedback on each driver. The period 1982–2015 is taken as reference for the observations, while models span 1956–2005. Maps show the mean from the ensemble of observations or four CMIP5 models: CCSM4, HadGEM2-ES, NorESM1-M, IPSL-CM5A-MR. Dotted pixels indicate a significant (p-value = 5%) strongest feedback agreed upon by at least 75% of the ensemble members.

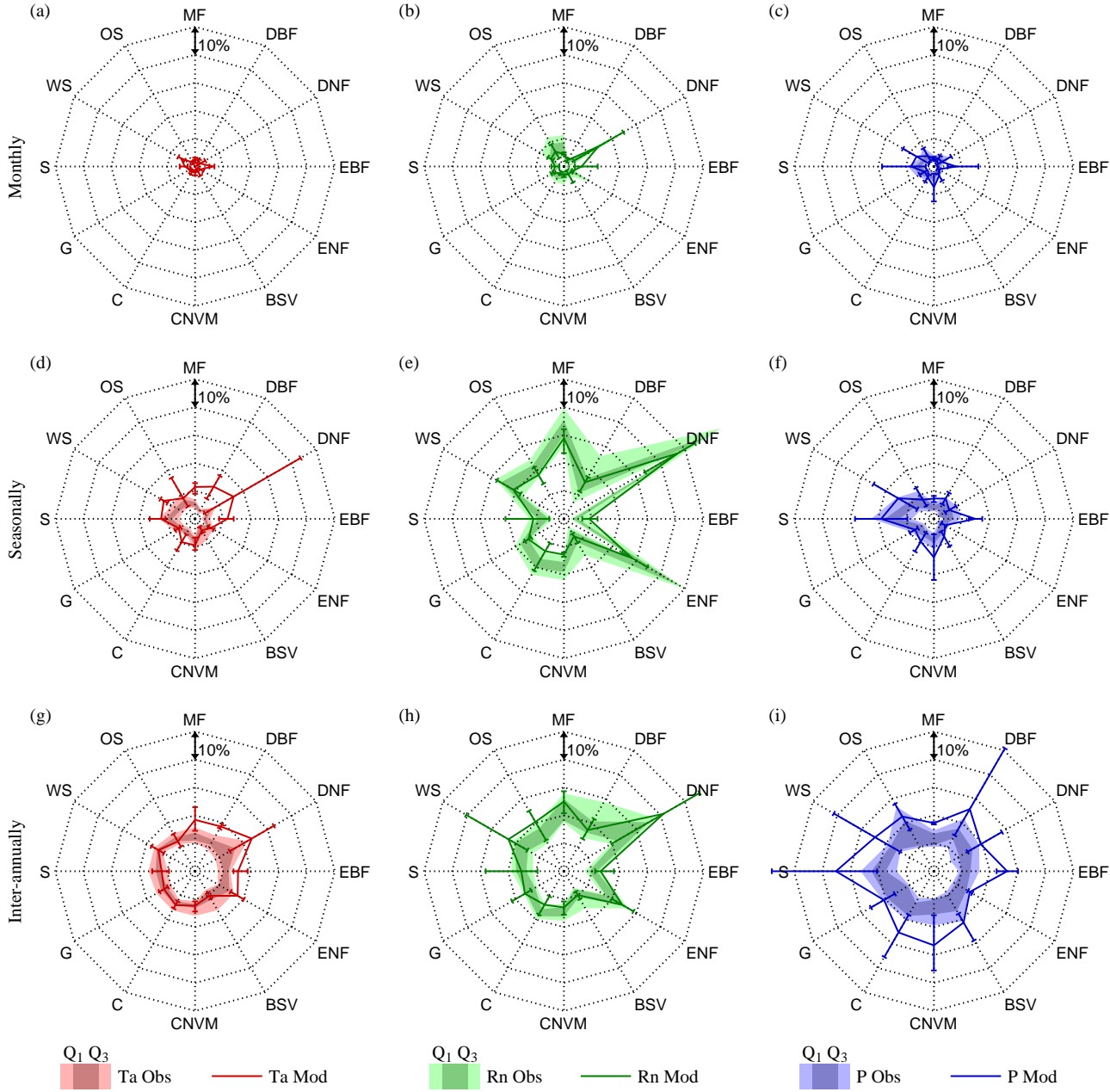

**Figure 4.** Climate impact on vegetation per biome. Biome averages of absolute observed (filled polygons) and modelled (lines) variation of LAI caused by air temperature (Ta), net radiation (Rn) and precipitation (P), at monthly **(a, b, c)**, seasonal **(d, e, f)**, and inter-annual **(g, h i)** time scales. Observations present the total range over all ensemble members, and the 25%- ($Q_1$) and 75%-percentile ($Q_3$). Models present an error-bar indicating the inter-model maximum, minimum and average results of four CMIP5 models (CCSM4, HadGEM2-ES, NorESM1-M, IPSL-CM5A-MR). Represented biomes are mixed forests (MF), deciduous broadleaf forest (DBF), deciduous needleleaf forest (DNF), evergreen broadleaf forest (EBF), evergreen needleleaf forest (ENF), barren or sparsely vegetated (BSV), cropland or natural vegetation mosaic (CNVM), croplands (C), grasslands (G), savannas (S), woody savannas (WS), and open shrublands (OS).

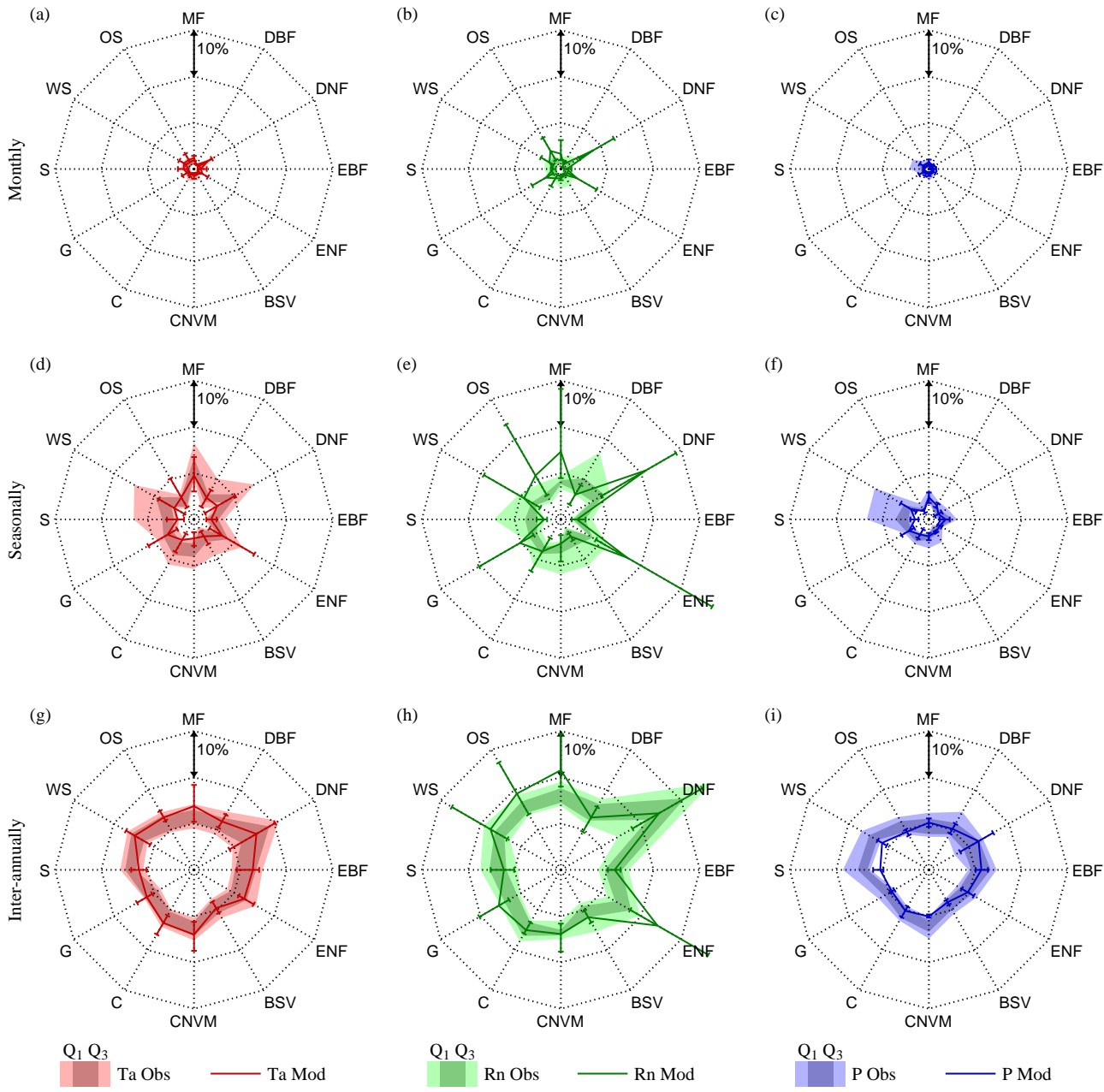

**Figure 5.** Vegetation feedback on climate per biome. Biome averages of absolute observed (filled polygons) and modelled (lines) variation of air temperature (Ta), net radiation (Rn) and precipitation (P) caused by LAI, at monthly **(a, b, c)**, seasonal **(d, e, f)**, and inter-annual **(g, h i)** time scales. Observations present the total range over all ensemble members, and the 25%- ($Q_1$) and 75%-percentile ($Q_3$). Models present an error-bar indicating the inter-model maximum, minimum and average results of four CMIP5 models (CCSM4, HadGEM2-ES, NorESM1-M, IPSL-CM5A-MR). Represented biomes are mixed forests (MF), deciduous broadleaf forest (DBF), deciduous needleleaf forest (DNF), evergreen broadleaf forest (EBF), evergreen needleleaf forest (ENF), barren or sparsely vegetated (BSV), cropland or natural vegetation mosaic (CNVM), croplands (C), grasslands (G), savannas (S), woody savannas (WS), and open shrublands (OS).

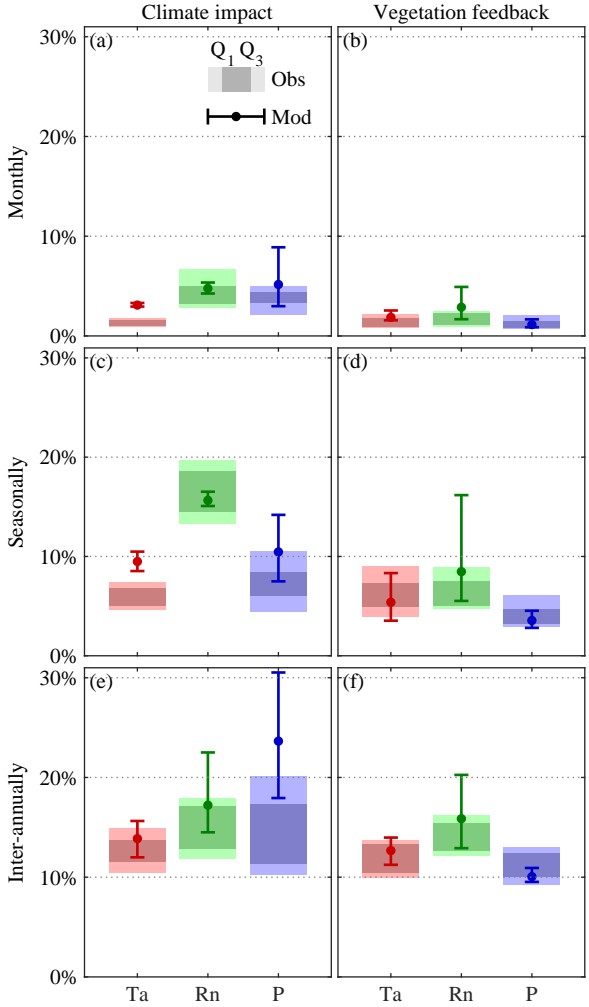

**Figure 6.** Global average climate impact on vegetation and vegetation feedback on climate. Global averages of absolute observed (filled rectangles), and modelled (lines) variation in vegetation (**a, c, e**) (climate (**b, d, f**)) caused by climate (vegetation), at monthly (**a, b**), seasonal (**c, d**), and inter-annual (**e, f**) time scales. Models present an error-bar indicating the inter-model maximum, minimum and average results of four CMIP5 models (CCSM4, HadGEM2-ES, NorESM1-M, IPSL-CM5A-MR).

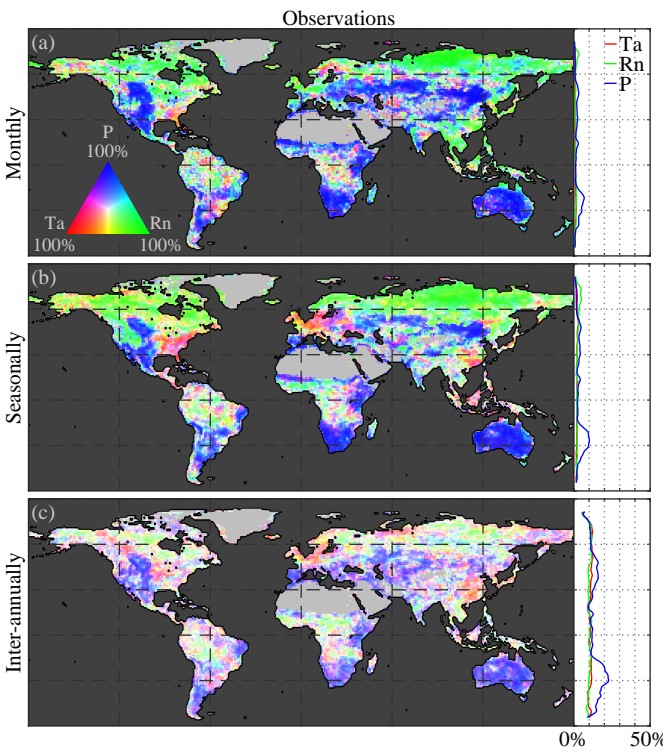

**Figure A1.** Global climate impact on anomalies of vegetation. Variability in observed anomalies of LAI caused by anomalies in air temperature (Ta), net radiation (Rn) and precipitation (P) at **(a)** monthly, **(b)** seasonal, and **(c)** inter-annual time scales. Maps show the causality in relative terms with respect to the dominant driver at each pixel, while the latitudinal profiles show the absolute impact of each driver. The period 1982–2015 is taken as reference for the observations.

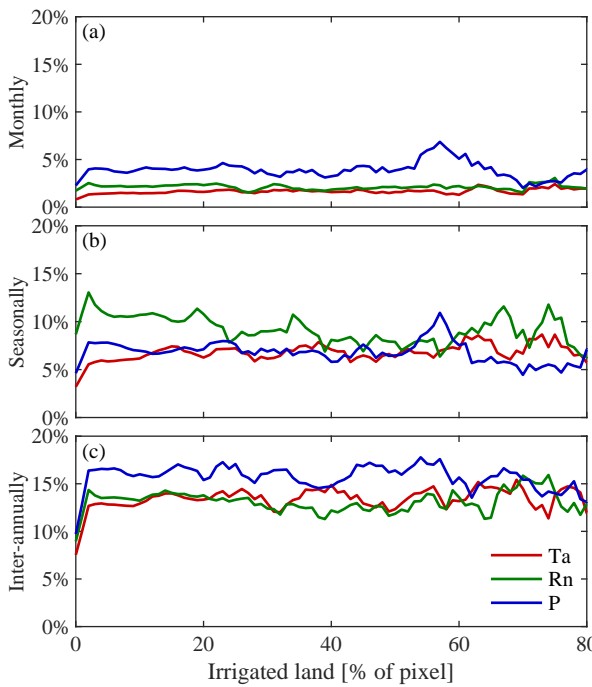

**Figure B1.** Impact of irrigation on the absolute explained variance in vegetation by climate. Variability in observed LAI caused by air temperature (Ta), net radiation (Rn) and precipitation (P) at **(a)** monthly, **(b)** seasonal, and **(c)** inter-annual time scales as a function of the area equipped for irrigation expressed as a percentage.

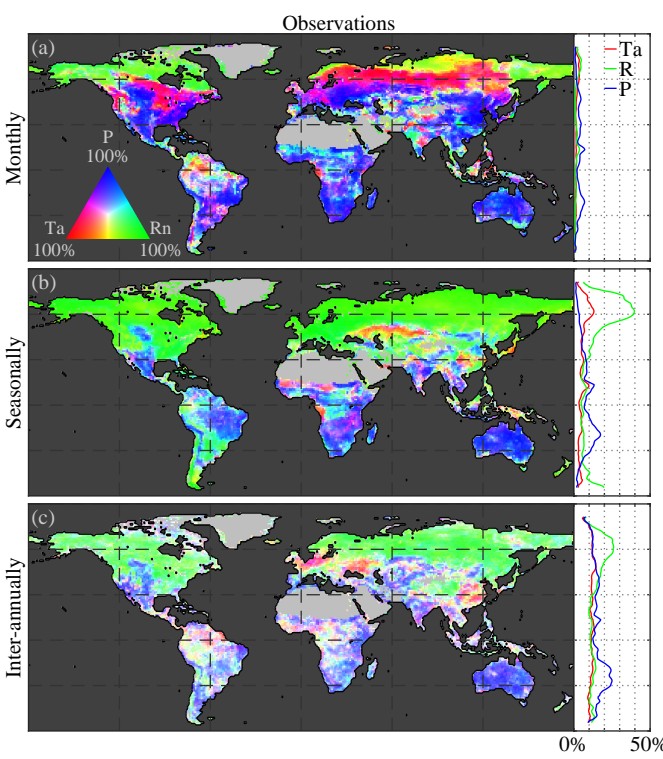

**Figure C1.** Global climate impact on vegetation using incoming radiation instead of net radiation. Variability in observed LAI caused by air temperature (Ta), incoming radiation (R) and precipitation (P) at **(a)** monthly, **(b)** seasonal, and **(c)** inter-annual time scales. Maps show the causality in relative terms with respect to the dominant driver at each pixel, while the latitudinal profiles show the absolute impact of each driver. The period 1982–2015 is taken as reference for the observations.

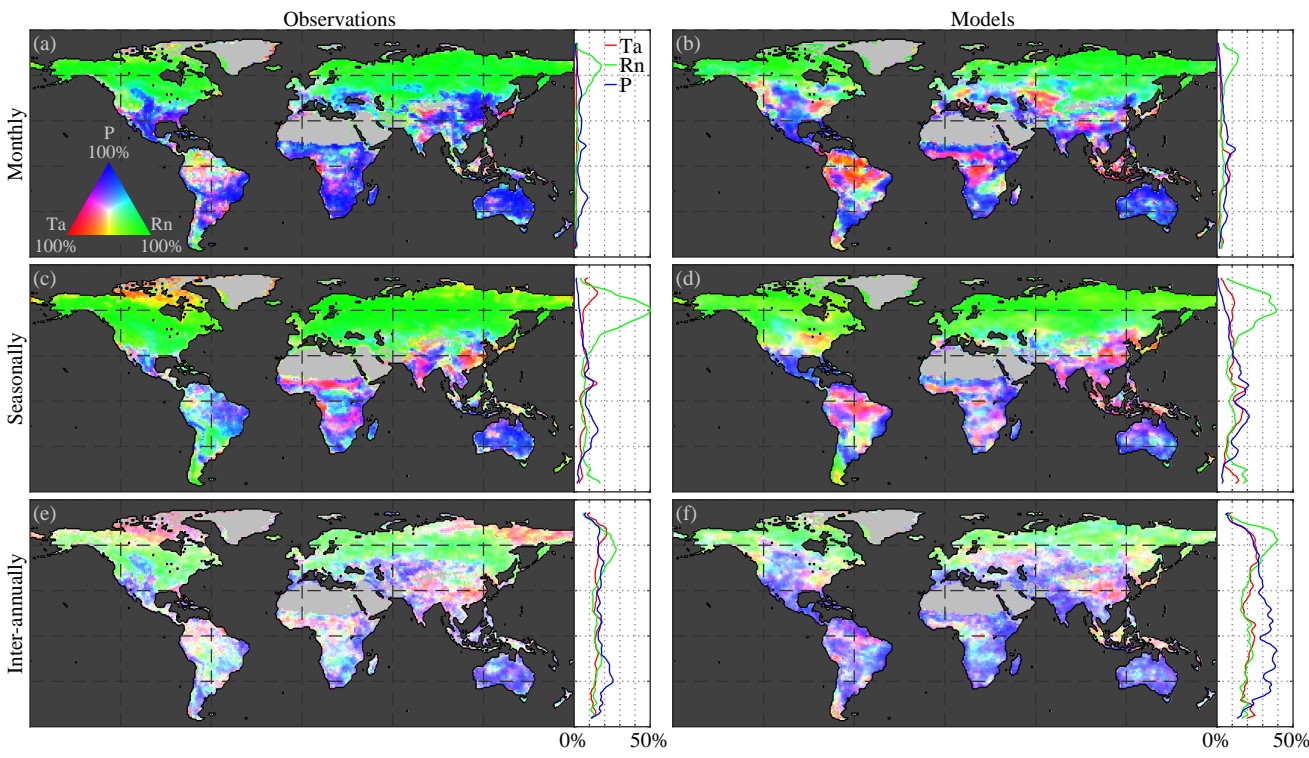

**Figure D1.** Global climate impact on vegetation during 1982–2005. Variability in **(a, c, e)** observed and **(b, d, f)** modelled LAI caused by air temperature (Ta), net radiation (Rn) and precipitation (P) at **(a, b)** monthly, **(c, d)** seasonal, and **(e, f)** inter-annual time scales. Maps show the causality in relative terms with respect to the dominant driver at each pixel, while the latitudinal profiles show the absolute impact of each driver. Maps show the mean from the ensemble of the observations or four CMIP5 models: CCSM4, HadGEM2-ES, NorESM1-M, IPSL-CM5A-MR.

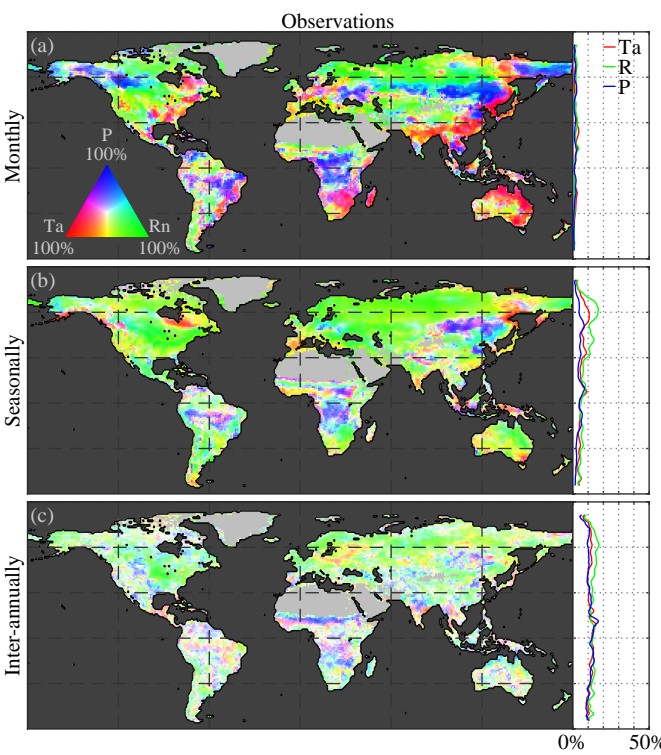

**Figure E1.** Global vegetation feedback on climate using incoming radiation instead of net radiation. Variability in air temperature (Ta), incoming radiation (R) and precipitation (P) that is caused by observed LAI at **(a)** monthly, **(b)** seasonal, and **(c)** inter-annual time scales. Maps show the causality in relative terms with respect to the strongest feedback at each pixel, while the latitudinal profiles show the absolute feedback on each driver. The period 1982–2015 is taken as reference for the observations.