# Peer review of "Global biosphere--climate interaction: a multi-temporal scale appraisal of observations and models"

_Biogeosciences, 2019_

## Referee Comment (RC1) · Anonymous Referee #1 · 8 Jul 2019

The authors used the Conditional Spectral Granger Causality framework to evaluate local biosphere – climate interactions at a global scale. This was done at three temporal scales (monthly, seasonaly and yearly) and using vegetation dynamics (based on LAI) retrieved from both satellite observations and earth system model data. Overall, I think it is a very interesting and innovative approach. Although the method has a few restrictions, such as the inability to account for off-site effects of vegetation on climate, these are well acknowledged in the discussion.

Comments: 1. The authors could consider to change the title to multi-temporal scales instead of scales as the latter may also refer to spatial scale. 2. Although not in the

scope of this paper, could the approach be useful to evaluate how the interactions change over time? 3. There is likely an important anthropogenic effect on both vegetation and climate dynamics. Could this impact the obtained results? 4. The interannual impact of climate on vegetation is also very patchy over Africa and North America in contrast to the modeled output (fig 2). Do the authors have an idea why this happens? Is it a methodological issue, data issue or are the drivers of long term trends more spatially heterogeneous (which is not catched by the models). 5. Did the authors try to run the analysis over the same time period for the remote sensing and model data (page 4, line 17)? Do the results substantially differ? 6. The approach includes data outside the growing season to estimate the monthly interactions. Yet, variations in LAI might not be meaningful during this period. Could this potentially affect the results? 7. What is the policy of the authors concerning sharing data/scripts? Are the authors planning to make these available via a repository/upon reasonable request/...?

---

## Referee Comment (RC2) · Anonymous Referee #2 · 11 Jul 2019

The manuscript explores the biosphere-climate interactions at global scale. The method, based on a Granger Causality framework, quantifies the climate impact on vegetation and the vegetation feedback on climate using satellite observations. The same approach is then applied to four ESMs and differences between data and model results are discussed. The study is well written and potentially interesting as – to my knowledge – is the first work aimed to isolate the climate-vegetation interactions analytically using observations and can help the modelling community to improve ESMs. However, I have some major concerns that need to be carefully addressed before publication.

Major comments 1) The study is based on a limited set of observational datasets: only one product per variable. In particular, LAI and precipitation data show large discrepancies and inconsistencies across products (Jiang et al., 2017). Results, based on a so limited set of products, may be largely affected by specific product uncertainties. The analysis should be replicated by using an ensemble of different products for LAI, P and possibly T and RN. Results based on an ensemble of combinations would be much more robust. Comparison of results obtained from different combinations of products would also enable you to assess the validity of your approach and the consistency of your results. Jiang, C. et al. Inconsistencies of interannual variability and trends in longterm satellite leaf area index products. Glob. Change Biol. 23, 4133-4146 (2017). 2) Spatial patterns shown in figures (e.g., figs. 2, 3 and appendices) are very jeopardized and - a part of the radiation control patterns - are not very credible. There is a huge spatial heterogeneity even in regions characterized by the same environmental conditions. I'm wondering, if such spatial variability reflects some problems of stability in the algorithm or noise in the modelled signal. These strange patterns emerge particularly at longer time scales (seasonal, interannual) maybe because the sample size is more limited (?). I really find difficult to believe in such patterns and authors should make an extra-effort to improve or at least understand such spatial variability. In my opinion, such spatial variability could originate from the native time series (possible uncertainties in the signal) and the processing of the signal, as I do not see any patterns that can be easily related to physical conditions. Maybe, the use of ensemble of different observational products (see comment 1) may help to retrieve a more robust signal. 3) The benchmark of ESMs is very useful and interesting. However, the authors should try to identify potential areas of model improvements. This exercise should be aimed to clearly understand what are the strengths and deficiencies of each single model with respect to the data-model comparison performed. A table to synthesize areas of improvements could help to convey the key information to modellers. 4) Remote sensing LAI data in winter season are affected by snow cover conditions. I'm wondering how you have addressed this issue. If you did not account for this, I think your results may

be strongly affected by this bias. 5) The relevance of the multi-temporal scale needs to be clarified, what is the added value of a such analysis compared to previous studies focusing only on monthly scale?

Specific comments Page 1 Paper of interest: https://www.nature.com/articles/s41467-019-10105-3 Line 9: Are you referring to the onset, end of growing season, or what? please, clarify. LAI is not synonym of phenology Line 11: For completeness, can you also briefly refer to the role of temperature? Line 13: please, specify over which temporal scale? Line 15: again, it is not clear here phenology to what is referring to? Line 17: I found a bit too much speculative the interpretation ... could not be just because the direct effect of climate on LAI is larger than the opposite feedback of vegetation on climate in nature? The fact that you are focusing on local scale without remote effect does not imply per se that the feedback of vegetation could be larger than the climate impact on vegetation...

Page 2 Line 8: our biosphere  $\rightarrow$  the biosphere Line 10: our Earth system  $\rightarrow$  the Earth system Line 12: you give per granted that models do not work well... I would reformulate the sentence... something like : model have shown limitations in capturing ... Line 14: Consider to include the following publication: https://www.earth-syst-scidata.net/10/1265/2018/ Line 15: Given its relevance in the article, I would contextualize briefly the multi-temporal issue already in this first paragraph. Line 15: Clarify why it is important "the representation of particular inter-variable sensitivities" Line 24: Please, clarify why is important to explore the multi-temporal issue. This would help the reader to follow your rationale and to better appreciate your findings. I would also stress here the challenges that you try to address. From what I understood the multi-temporal scales and the explicit representation of causal relation between vegetation and climate represent the key novelty of your work. I would put more emphasis on these two aspects. Line 25: I would mention that Papagiannopoulou et al. do not address the seasonal and inter-annual scales in order to clearly differentiate your study form the previous work.

СЗ

Page 3 Line 1: I would suggest integrating your literature review with these relevant articles. https://science.sciencemag.org/content/351/6273/600 https://www.nature.com/articles/s41467-017-02810-8 Line 18: in principle, it may serve also to detect areas where models work well. I would rephrase a bit the sentence in a more general way. Line 31: remote sensing LAI data in winter season are affected by snow cover conditions. I'm wondering how you have addressed this issue. If you did not account for this, I think your results may be strongly affected by this bias.

Page 4 Line 4: Why you do not use the ESA-CCI land cover product (and conversion to pass to PFT)? In principle this would enable to track for changes in PFT over almost the entire period of study. The ESA-CCI product represent the state-of-the-art product aimed to improve the link between remote sensing users and climate modelers ... https://www.esa-landcover-cci.org/ Line 8: Given the large differences amongst different products for some of the variables considered, I would strongly suggest to account for multiple products (https://onlinelibrary.wiley.com/doi/10.1111/gcb.13787). For instance, for LAI, data from GLASS, LTDR, GLOBMAP could also be included in the study. The same for precipitation which show large discrepancies - especially at interannual scale - depending on the dataset used. The use of ensemble of observational products would make your results more robust and substantially improve the work. Line 9: Please, clarify the value of using online model simulations in place of offline simulations. I see a potential limitation as in online ESMs the climate signal may largely determine the response of the land surface and then mask the interplay between vegetation and biophysical processes. Further reading: Blyth, E., Clark, D. B., Ellis, R., Huntingford, C., Los, S., Pryor, M., et al. (2011). A comprehensive set of benchmark tests for a land surface model of simultaneous fluxes of water and carbon at both the global and seasonal scale. Geoscientific Model Development, 4(2), 255-269. https://doi.org/10.5194/gmd-4-255-2011 Winckler, J., Reick, C. H., & Pongratz, J. (2016). Robust identification of local biogeophysical effects of landaĂŘcover change in a global climate model. Journal of Climate, 30(3), 11591176. https://doi.org/10.1175/JCLI-D-16-0067.1 Line 13: Please, clarify the selection, why only these 4 models are used here? Line 15: Are all models run under a consistent modelling setup (e.g., same land cover changes, same climate forcing)? Please, clarify. The consistency in modelling experiment is important to compare the different model results each other. Line 17: Not sure this is correct. You basically used two different periods of analysis for observations and models: 1981-2015 (ca 35 years) for observations; 1956-2005 (50 years) for models. A part of the temporal shift between the two experiments, I would suggest to verify that the different length in the time series do not introduce a systematic bias between observational- and model-based results. Why you decided to start from 1956 for models? To me it would be more logic at least th preserve the same length of observations (35 years). Please, check this and clarify your choices.

Page 5 Line 24: In the presented formulation of GC, the temporal lag m is implicitly assumed the same for all predictors. In practice, I expected that the legacy effects may differ depending on the predictor. Can this be included in the formulation? Please, discuss the implications. Line 27: In principle, data could be aggregated at seasonal or annual level and the GC applied to such values. I presume that the limited sample size hampers the use of GC in a such "aggregated" mode. Please, clarify.

Page 6 Line 2: temporal scale-dependent Line 19: I believe that eq. 7 needs more clarification for readers not familiar with the method. Line 29: Zp in place of Xp?

Page 7 Line 14: I would clearly mention that CSGC does not allow to quantify the sign of causal relation. It is already mentioned in results... but I would also mention here - or somewhere in the method section - because important.

Page 9 Line 3: I would refer to seasonal LAI variability here and in the rest of the manuscript. Phenology implies other metrics that are not accounted for in this work Line 8: please clarify this upper value

Page 10 Line 9: Could irrigation or land enlargement, particularly relevant in some

regions of the globe, partially explain some patterns (e.g., India and China)? Should not be the irrigated lands factored out? https://www.nature.com/articles/s41893-019-0220-7 Line 16: I presume that if you mask irrigated lands this fraction will increase. Can you please comment on this. Line 17: same detrending and deseasonalization approach used for predictors, right? Line 22: compared to what? Fig A1? PA17? Line 23: The snow precipitation should not be already considered in CRU data that used here? Line 24: battery  $\rightarrow$  set? Line 22: also the methods used to quantify the causal relations differ

Page 11 Line 1: forcing on vegetation Line 6: I find the dominance of precipitation very elusive... There are not clear patterns emerging at interannual scale. Probably the P control is just a bit over the other drivers ... but to me what emerges from figure 2e is a major co-dominance of multiple drivers. Please, can you please comment on this. Line 28: To me the comparison performed only on these numbers is misleading because they refer to the relative contribution to the total explained variance. Therefore, ESMs could be in principle represent well the variability of the T control on vegetation in absolute terms, but could overestimate the P control on vegetation in absolute terms. This would lead to an underestimation of the T control in relative terms over the globe ... again not because they fail to represent the T control but because they fail the P or RN controls. The analyses should be complemented with the comparison in absolute terms.

Page 12 Line 1: clarify what do you mean? precipitation and temperature? Line 20: I found the patterns from observations very jeopardized across all temporal scale analysed, and in particular at seasonal and interannual scales. It is really difficult to believe that in the real word you pass from one dominant control to another one while remaining in the same environmental conditions. This heterogeneity should be better explored and understood. The use of ensemble of combination of different observational product of LAI, P, T, RN could help to derive more robust and spatially consistent patterns.

Page 13 Line 13: clearer Line 19: You could move figure 6 earlier and refer to it. Line

25: ESMs capture correctly the LAI effects on net radiation throughout most of the Northern Hemisphere. How do you reconcile with results from Forzieri et al., 2018? Forzieri, G., Duveiller, G., Georgievski, G., Li, W., Robertson, E., Kautz, M., Lawrence, P., Garcia San Martin, L., Anthoni, P., Ciais, P., et al.: Evaluating the Interplay Between Biophysical Processes and Leaf Area Changes in Land Surface Models, Journal of Advances in Modeling Earth Systems, 2018.

Page 14 Line 9: I would say only for EBF, DNF, DBF, MF. For the rest of classes the data-model comparison is fine... Line 19: Only when averaged at biome level. Maps in figure 3 differ substantially. Maybe this concept would merit to be expanded a bit. Results from ESMs and satellite tend to converge when averaged at biome level... can you please comment on this? Line 30: Can you please reconcile or at least interpret these divergences?

Figure 2, 3 and appendices: I suggest a different colour palette because colours tend to saturate quickly and differences cannot be appreciated well. Figure 4: change order of variables consistently with figure legend. Same f

---

## Author Comment (AC1) · 25 Jul 2019

Thank you for your insightful comments and support of the manuscript. Please find attached all your comments and our responses.

Please also note the supplement to this comment:
https://www.biogeosciences-discuss.net/bg-2019-212/bg-2019-212-AC1-supplement.pdf

---

## Author Comment (AC2) · 25 Jul 2019

We thank the reviewer for the insightful comments and we hope that we have addressed them all adequately. We listed your comments and our responses in the attached supplement.

Please also note the supplement to this comment:
https://www.biogeosciences-discuss.net/bg-2019-212/bg-2019-212-AC2-supplement.pdf

---

## Author Response (AR1)

Here, an overview is provided of all replies to the comments of the anonymous reviewers. Each concern is tackled separately. First the comment is listed, and our reply is pasted underneath in italics. Also, if we adapted the manuscript, this is added to our reply (underlined sentences represent the changes made to the draft.

Directly after the responses to all comments, we included a version of the manuscript marking all the performed changes.

**Reply comments Reviewer #1**

 The authors used the Conditional Spectral Granger Causality framework to evaluate local biosphere – climate interactions at a global scale. This was done at three temporal scales (monthly, seasonally and yearly) and using vegetation dynamics (based on LAI) retrieved from both satellite observations and earth system model data. Overall, I think it is a very interesting and innovative approach. Although the method has a few restrictions, such as the inability to account for off-site effects of vegetation on climate, these are well acknowledged in the discussion.

Thank you for your insightful comments and support of the manuscript.

2. The authors could consider to change the title to multi-temporal scales instead of scales as the latter may also refer to spatial scale.

We acknowledge the possibility for confusion and have changed the title accordingly.

New title: Global biosphere-climate interaction: a multi-temporal scale appraisal of observations and models

3. Although not in the scope of this paper, could the approach be useful to evaluate how the interactions change over time?

Conditional Spectral Granger Causality can be used to address changes over time (see Dhamala et al., 2008). Currently, the wavelet spectrum is averaged over all time steps. This results in more robust statistical measures (i.e. higher statistical confidence), at the expense of a partial loss of time domain information. Resolving the time-frequency space requires more statistical power, which can only be achieved by averaging across large spatial extent. As the reviewer states, exploring the time dimension lies out of the scope of the current manuscript, but is definitely part of our future plans.

Dhamala, M., Rangarajan, G. & Ding, M. (2018). Estimating Granger causality from Fourier and wavelet transforms of time series data. Phys. Rev. Lett., 100, 018701.

4. There is likely an important anthropogenic effect on both vegetation and climate dynamics. Could this impact the obtained results?

Anthropogenic effects impacting climate on the long term, and resulting in multi-decadal trends, are not addressed in this study due to the limited data record. As the reviewer noticed, based on the three-decade record length available for the analyses, and considering unavoidable time series edge effects, we can only

resolve reliably time scales with periods smaller than 10 year. Conversely, vegetation disturbances directly driven by human activities, such as deforestation, agricultural practices, etc., can generate variability in the vegetation indices on a much broader range of scales, thus directly affecting our analyses. However, because this variability is not necessarily related to climate, it will be assimilated into residual noise in our approach. Granger causality assumes causal sufficiency and regions with unobservable causes (in this case those leading to e.g. deforestation) will be poorly resolved by the framework. This is now explicitly mentioned in the revised version.

In text, Sect. 3.1: Noteworthy is that anthropogenic effects, which are not addressed here, can also impact vegetation and climate at short temporal scales. For example, irrigation and deforestation can result in a decoupling between climate and vegetation (Lawrence et al., 2015; Chen et al., 2019). In the tropics, deforestation results in a warming effect due to reduced plant transpiration, which in turn may induce a decline in precipitation, creating a warmer and drier regime (Lawrence et al., 2015). Irrigation allows for growing crops in water-limited regions, consequently inducing energy constraints which are captured by the CSGC. Note that due to the limited data record, the effects of global warming trends and carbon dioxide fertilisation – and the consequent trends in vegetation greening and water use efficiency (Reichstein et al., 2013; Wu et al., 2015; Zhu et al., 2016) – are not directly addressed in this study.

5. The inter-annual impact of climate on vegetation is very patch over Africa and North America in contrast to the modelled output (fig 2). Do the authors have an idea why this happens? Is this a methodological issue, data issue or are the drivers of long term trends more spatially heterogeneous (which is not catched by the models).

We thank the reviewer for this comment. We further explored these patterns after the referee's comment, and concluded that the shape parameter of the Morlet wavelet partly influences the results in this regard. This parameter provides a trade-off between spectral and temporal resolution. By increasing the timeresolution the conditional causality patterns at inter-annual scales can be better resolved. This improves the clarity in the figures, even if some heterogeneity remains. A further improvement followed one of the comments of Reviewer #2 (see comment #2), who requested an ensemble based on multiple datasets of LAI and climate variables, which would help resolve issues related data errors. Both the tuning of the shape parameter as well as the creation of the ensemble are adopted in the revised version and all figures are updated accordingly. We described the selection of the frequency parameter to balance the time and frequency resolution in the manuscript.

In text, Sect 2.2.2: ...between predictors and target variable. In order to perform the time-frequency decomposition, the Morlet wavelet is used and a balance between the time and frequency resolutions is obtained by setting the shape parameter to a value of 6, as in Torrence and Compo (1998), or Casagrande et al. (2015). Moreover, to overcome the limitation...

6. Did the authors try to run the analysis over the same time period for the remote sensing and model data (page 4, line 17)? Do the results substantially differ?

We thank the reviewer for this comment. We have already run the analysis over the same time period and found that almost no changes occurred. The results for inter-annual scales faint slightly, but the major patterns remain consistent. We have included the results to the supplementary and discussed them in the main text.

In text, Sect. 2.1.2: ...under the assumption that sensitivities are stationary (see e.g. Green et al., 2017). Sect. 3.2 addresses the validity of this assumption. Nonetheless, we...

In text, Sect. 3.2: ... (Fig 2e), and is also strongly overestimated in absolute terms at most latitudes, especially in the tropics. Further analysis shows that the divergence in the considered period between observations and models (see Sect. 2.1) does not substantially impact results; repeating the analysis for the overlapping time range for observations and models (1982–2005) yields very similar findings (Fig. C1).

7. The approach includes data outside the growing season to estimate the monthly interactions. Yet, variations in LAI might not be meaningful during this period. Could this potentially affect the results?

Yes, we agree with the reviewer that it would be more meaningful from a biological point of view to include only data from the growing season. We are currently working to resolve the time domain to the Conditional Spectral Granger Causality formulation (see comment #3), which would allow us to tackle this issue explicitly in the future. Below you can find a preliminary figure that shows the percentage of explained variation by climate at a monthly scale over a 10-year period for a pixel located in central Russia calculated using CSGC to address changes over time. We can clearly see that during wintertime, air temperature (and radiation) generally seem to inhibit vegetation growth, while during summer, when the temperature is high enough for plants to grow, water limitation spikes. This figure shows how we can disentangle drivers from in- and outside the growing season as they do differ. However, as stated in comment #3, this is out of the scope of this paper.

Also, we are confident that the adoption of an ensemble approach (see Reviewer #2 comment #2) will dampen the sensitivity of our method to the errors in the individual data sources, thus removing product-specific biases in wintertime.

8. What is the policy of the authors concerning sharing data/scripts? Are the authors planning to make these available via a repository/upon reasonable request/...?

We are open for sharing the scripts using GitHub after publication at https://github.com/lhwm. All datasets used in this study are freely available, for which links will be provided in the README.md file of the GitHubpage.

**Reply comments Reviewer #2**

 The manuscript explores the biosphere-climate interactions at global scale. The method, based on a Granger Causality framework, quantifies the climate impact on vegetation and the vegetation feedback on climate using satellite observations. The same approach is then applied to four ESMs and differences between data and model results are discussed. The study is well written and potentially interesting as – to my knowledge – is the first work aimed to isolate the climate-vegetation interactions analytically using observations and can help the modelling community to improve ESMs. However, I have some major concerns that need to be carefully addressed before publication.

We thank the reviewer for the insightful comments and we hope that we have addressed all major and minor concerns adequately.

**General comments**

2. The study is based on a limited set of observational datasets: only one product per variable. In particular, LAI and precipitation data show large discrepancies and inconsistencies across products (Jiang et al., 2017). Results, based on such a limited set of products, may be largely affected by specific product uncertainties. The analysis should be replicated by using an ensemble of different products for LAI, P and possibly T and Rn. Results based on an ensemble of combinations would be much more robust. Comparison of results obtained from different combinations of products would also enable you to assess the validity of your approach and the consistency of your results. (Jiang, C. et al. Inconsistencies of interannual variability and trends in longterm satellite leaf area index products. Glob. Change Biol. 23, 4133–4146 (2017).)

We thank the reviewer for this comment. We agree that by creating an ensemble of LAI, P, T and Rn, we can significantly improve the robustness of the results. We have added three more LAI products, corresponding to those used in Jiang et al. (2017). Furthermore, for climate we added air temperature and net radiation from ECMWF's most recent reanalysis product ERA5, and two more precipitation products, namely ERA5 and GPCC. We added more products for LAI and precipitation as for air temperature or net radiation due to the larger inter-product variability. A brief description of all included products was added to the data section, and figures were updated after repeating the analysis for all ensemble members. Results for all model ensembles generally agree well (see comment 3).

In text, Sect. 2.1.1: To avoid product-specific biases and artefacts, an ensemble of multiple observation-based products for each variable is created, consisting of: (a) four LAI, (b) two air temperature, (c) two net radiation, and (d) three precipitation data sets. The larger ensemble of data sets here adopted to characterise LAI and precipitation is motivated by the larger disparity among the different products of these variables (Jiang et al., 2017; Sun et al., 2018). Tab. 1 provides an overview of the available datasets. Finally, the International Geosphere-Biosphere Program (IGBP) land cover classification (Loveland and Belward, 1997) is used to determine biome-specific behaviours...

| Table 1. Summary of global data sets used for vegetation, i.e. LAI, and climate, i.e. air temperature (Ta), net radiation (Rn), and |
|-------------------------------------------------------------------------------------------------------------------------------------|
| precipitation (P).                                                                                                                  |

| Draduat                                                        | Minimum resolution   |                        | Variable          | Deferreres              |
|----------------------------------------------------------------|----------------------|------------------------|-------------------|-------------------------|
| Product                                                        | Spatial              | Temporal               | variable          | Rejerenco               |
| Global Inventory Modelling and Mapping Studies 3 rd | 1/170                | 1982-2015;             |                   | 7hu at al (2012)        |
| generation (GIMMS3g)                                           | 1/12                 | bimonthly              | LAI               | Zhu et ul. (2015)       |
| NOAA/AVHRR Thematic Climate Data Record                        | 0 05°                | 5° 1982-2018;
daily |                   | Claverie et al. (2016)  |
| (TCDR) Reflectance                                             | 0.05                 |                        | LAI               | claverie et al. (2010)  |
| GIMMS3g + Terra/MODIS C5 reflectance                           | 1/12 75° 1982-2017;  |                        | Liu et al. (2012) |                         |
| (GLOBMAP)                                                      | 1/15.75              | 28-day                 | LAI               |                         |
| NOAA/AVHRR LTDR + Terra/MODIS C5 reflectance                   | 0.05°                | 1982-2015;             |                   | Xiao et al. (2016)      |
| (GLASS)                                                        |                      | 8-day                  | LAI               | Xido et di. (2010)      |
| ECMM/E ERAS                                                    | 32km 1979;
hourly | 1979;                  | Ta, Rn and P      | Hersbach and Dee (2016) |
|                                                                |                      | JZKIII                 |                   | nersbach and Dee (2010) |
| Climate Research Unit – National Centers for                   | 0.05° 1:
6-       | 1901-2016;             | Ta, Rn and P      | Viouv (2018)            |
| Environmental Prediction (CRU-NCEP) version 7                  |                      | 6-hour                 |                   | viový (2018)            |
| Slobal Precipitation Climatology Centre (GPCC)                 | 0 5°                 | 1891-2016;             | D                 | Schneider et al. (2011) |
|                                                                | 0.5                  | daily                  | i          |                         |

3. Spatial patterns shown in figures (e.g., figs. 2, 3 and appendices) are very jeopardized and – a part of the radiation control patterns – are not very credible. There is a huge spatial heterogeneity even in regions characterized by the same environmental conditions. I'm wondering, if such spatial variability reflects some problems of stability in the algorithm or noise in the modelled signal. These strange patterns emerge particularly at longer time scales (seasonal, inter-annual) maybe because the sample size is more limited? I really find difficult to believe in such patterns and authors should make an extra effort to improve or at least understand such spatial variability. In my opinion, such spatial variability could originate from the native time series (possible uncertainties in the signal) and the processing of the signal, as I do not see any patterns that can be easily related to physical conditions. Maybe, the use of ensemble of different observational products (see previous comment) may help to retrieve a more robust signal.

We are aware of the heterogeneity at longer timescales, and also reviewer #1 (see comment #5) pointed to this issue. The problem is partly due to the parametrization of the frequency parameter of the wavelet, which provides a trade-off between temporal and spectral resolution. As mentioned in the response to reviewer #1 (see comment #5), fine-tuning the parameter to increase temporal resolution can improve the inter-annual patterns. However, the noisy patterns can also occur due to noise in the input data, as mentioned in comment #2, and the length of the available sample. By creating an ensemble of datasets and changing the frequency parameter, the noise related to errors in the data is greatly reduced, resulting in more homogeneous patterns, even at inter-annual scales. All figures are updated, and the result section is updated accordingly (only minor changes). We described the selection of the frequency parameter to balance the time and frequency resolution (see Reviewer #1 comment #5).

4. The benchmark of ESMs is very useful and interesting. However, the authors should try to identify potential areas of model improvements. This exercise should aim to clearly understand what are the strengths and deficiencies of each single model with respect to the data-model comparison performed. A table to synthesize areas of improvements could help to convey the key information to modelers.

Although we agree this might be of interest to the modelling community, we aimed not to single out individual models in this manuscript due to length restrictions. Moreover, we believe that by focusing on model

differences, or even specific model parameterisations, we might dilute the main findings that relate to the whole range of models. Therefore, we believe a model-specific interpretation is outside the scope of this study, and hope the reviewer may agree with this rationale.

5. Remote sensing LAI data in winter season are affected by snow cover conditions. I'm wondering how you have addressed this issue. If you did not account for this, I think your results may be strongly affected by this bias.

We acknowledge that snow cover might affect the LAI in the high northern latitudes, especially at the seasonal and monthly scales. At the moment, we do not address this issue. As pointed in the response to reviewer #1 (see comment #7), we are currently working on adapting the CSGC algorithm to explicitly resolve different time steps/periods, which would, in the future, allow to resolve the causal relationships in time and mask out periods of poor data quality. This however requires an in-depth adaptation of the method. In the response to reviewer #1 (see comment #7), we added a preliminary figure showing the temporal variation in explained variance by three drivers over a 10-year period. This clearly shows that, in the future, we can tackle this issue. Furthermore, we are also confident that the adoption of an ensemble approach (as proposed by comment #2) will dampen the sensitivity to errors in individual products, during for instance wintertime, being however aware of the fact that these errors are likely systematic and shared by all data products. In the revised manuscript, this issue is explicitly discussed.

In text, Sect. 3.1: ...incoming radiation during winter months. However, in those latitudes, LAI retrievals are contaminated by snow cover signals. While focusing on the growing season could solve this issue, the CSGC requires continuous time series. Because in wintertime, due to limitations in solar radiation, plant growth is inhibited in northern latitudes, most variability captured at monthly scales will be dominated by the more dynamic spring and summer periods; therefore, our results suggests that radiation still dominates the behaviour of vegetation at these latitudes. This dominant high-latitude radiation control was not reported by Papagiannopoulou et al. (2017b), who, based on a non-linear Granger causality framework, found that 61% of the vegetated land...

6. The relevance of the multi-temporal scale needs to be clarified, what is the added value of such analysis compared to previous studies focusing only on monthly scale.

We feel that the fact our results differ for different frequencies (time-scales) highlights by itself the need to consider these frequencies separately to better understand the driving role of climate in ecosystem dynamics. To better clarify this point, we revised the manuscript to explain why, conceptually, phenology scales and inter-annual variability also need to be considered separately when models are evaluated.

In text, Sect. 1: ... Despite efforts to identify the controls of vegetation, which showed that ESMs overestimate the annual LAI due to problems related to the timing of the phenological cycle (Anav et al., 2013; Murray-Tortarolo et al., 2013; Zhu et al., 2013; Forkel et al., 2014; Verger et al., 2016), ESMs remain in need of a better understanding of why these multi-temporal scale variabilities differ from the observations. *Rather than using correlation or regression techniques, a method capable of inferring causality can greatly aid our understanding of key climate-biosphere processes, which in turn can help enhance the ESMs (Runge et al., 2019).* In a recent example,...

**Specific comments:**

**Page 1**

• Paper of interest: https://www.nature.com/articles/s41467-019-10105-3 (Runge et al.: Inferring causation from time series in Earth system sciences, Nature communications, 10.1, 2553, 2019.).

At the time of submitting the first version of the manuscript, this article was not officially published yet, despite it being of interest for our manuscript. We've added it to the manuscript (see reply comment #6).

• Line 9: Are you referring to the onset, end of growing season, or what? Please clarify. LAI is not synonym of phenology.

As this is a recurring comment, we will clarify here what we mean by 'phenology'. We are aware that LAI seasonality is not a synonym of phenology, however, we do believe that LAI is highly sensitive to phenological changes, as also shown by Richardson et al. (2009) and Verger et al., 2016, just as Vegetation Optical Depth (VOD) is often used as a proxy of the vegetation total biomass due to its close proximity to the vegetation water content Liu et al. (2011). Here, if we speak of phenology, or phenological cycle, we mean the dynamics of LAI over the seasons. If CSGC points towards a certain climate control on vegetation at seasonal scale, this means that the variation in climate within a year, i.e. seasonality, contains information that can explain the behaviour of vegetation, but these two are closely linked. One can state than that the total phenological cycle, e.g. the timing (start and end) of the growing season, the amplitude of the variations, etc. are strongly forced by radiation in this example. We incorporated this definition in the main manuscript, as we feel the abstract is not the right place to do so. (References not in manuscript: Liu et al. Global long-term passive microwave satellite-based retrievals of vegetation optical depth, Geophysical Research Letters, 38, L18402, 2011).

In text, Sect. 2.2.6: ...differ from the short-term processes. Hereafter, the terms phenology and phenological cycle are used to refer to the seasonal-scale variability in LAI. This reflects features such as the timing of the growing season or the amplitude of the intra-annual cycle (Richardson et al., 2009; Verger et al., 2016), since CSGC will react to variability in both the time and frequency domains. As explained in Sect. 2.2.3, CSGC allows a simultaneous analysis of the interactions at multi-temporal scales, while no assumption...

In text, Abstract: ...The seasonal LAI variability in energy-driven latitudes...

• Line 11: For completeness, can you also briefly refer to the role of temperature?

In text: ... inter-annual than multi-month scales. Globally, precipitation is the most dominant driver of vegetation at monthly scales, particularly in (semi-)arid regions. The seasonal LAI variability in energy-driven latitudes is mainly controlled by radiation, while air temperature controls vegetation growth and decay in the high northern latitudes at inter-annual scales. The observational results...

• Line 13: Please, specify over which temporal scale?

In text: ... semi-arid regions at inter-annual scales. Analogously,...

• Line 15: Again, it is not clear here to what phenology is referring to?

We refer to the comment of page 1, line 9 that tackled this issue and hope it helped to solve the confusion.

In text: ... control of air temperature on seasonal forest variability. Overall...

• Line 17: I found a bit too much speculative the interpretation... Could not be just because the direct effect of climate on LAI is larger than the opposite feedback of vegetation on climate in nature? The fact that you are focusing on local scale without remote effect does not imply per se that the feedback of vegetation could be larger than the climate impact on vegetation.

We thank the reviewer for this remark. We did not mean to state that the feedback on climate is smaller as the climate impact on vegetation due to the local nature of our analysis. However, we do see how this statement could be interpreted this way. Therefore, we altered the statement slightly.

In text: ...Overall, climate impacts on LAI are found to be stronger than the feedbacks of LAI on climate in both observations and models; in other words, local climate variability leaves a larger imprint on temporal LAI dynamics than vice versa. Note however that while vegetation reacts directly to its local climate conditions, its dynamics may affect climate preferably downwind, especially in the case of precipitation. Consequently, the local (i.e. spatially collocated) character of the analysis does not allow for the identification of downwind or remote feedbacks, biophysical effects of vegetation on climate might be underestimated. Nonetheless, the widespread...

**Page 2**

• Line 8 and 10: our biosphere  $\rightarrow$  the biosphere, our Earth system  $\rightarrow$  the Earth system

Agree, corrected in text.

• Line 12: You give per granted that models do not work well... I would reformulate the sentence... something like: models have shown limitations in capturing...

Agree, the statement was open to interpretation. We changed it in the text.

In text: The different approaches to objectively evaluate the skill of Earth System Models (ESMs) in representing the two-way coupling between vegetation and climate have revealed several limitations (Randerson et al., 2009; ...

Line 14: Consider to include the following publication: https://www.earth-syst-sci-data.net/10/1265/2018/ (Duveiller et al., Biophysics and vegetation cover change: a process-based evaluation framework for confronting land surface models with satellite observations, Earth System Science Data, 10, 1265-1279, 2018)

We thank the reviewer for this suggestion and added the article to the references.

• Line 15: Clarify why it is important "the representation of particular inter-variable sensitivities"

We agree we left this open for interpretation. It is important to study these links between variables in models to identify which processes are missing, or remain under-represented. In addition, ESMs have been shown to overestimate the mean annual LAI due to overestimation of the length of the growing season (see comment #6).

In text: ...Most of these efforts focus on the evaluation of the magnitude and short-term dynamics of individual variables (such as LAI, and gross primary production, GPP), rather than on the inter-variable sensitivities, which would be more informative on whether the interplay between vegetation and climate is reliably represented in these models. Furthermore, previous benchmark studies have typically focused on one specific time scale (typically annually or monthly), while the ecosystem response to (and feedback on) climate is expected to vary for different time scales; e.g. a model may accurately replicate the observed interplay between vegetation and climate at monthly scales, but still fail to capture the sensitivities that become relevant at seasonal or inter-annual time scales.

 Line 24: Please, clarify why it is important to explore the multi-temporal issue. This would help the reader to follow your rationale and to better appreciate your findings. I would also stress here the challenges that you try to address. From what I understood, the multi-temporal scales and the explicit representation of causal relation between vegetation and climate represent the key novelty of your work. I would put more emphasis on these two aspects.

We thank the reviewer for this suggestion. We can see how the manuscript can benefit from stressing this more explicitly. We made some alterations in the text (see comment #6 and comment page 1, line 15).

• Line 25: I would mention that Papagiannopoulou et al. (2017b) do not address the seasonal and interannual scales in order to clearly differentiate your study from the previous work.

In text: In a recent example, Papagiannopoulou et al. (2017a,b) focused on evaluating multi-month vegetation variability in response to local climate, using a non-linear Granger causality framework applied to optical remote sensing indices. They showed that water availability and precipitation patterns primarily drive vegetation anomalies at monthly scales in more than 60% of the vegetated land, but did not address the relevant drivers over longer time scales. The inter-annual variability in terrestrial carbon fluxes has...

**Page 3**

 Line 1: I would suggest integrating your literature review with these relevant articles. https://science.sciencemag.org/content/351/6273/600 (Ramdane et al., Biophysical climate impacts of recent changes in global forest cover, Science, 351(6273), 600-604, 2016), https://www.nature.com/articles/s41467-017-02810-8 (Duveiller et al., The mark of vegetation change on Earth's surface energy balance, 9, 679, 2018)

We thank the reviewer for this suggestion and added the references to the literature review.

In text, Sect. 1: ... mainly due to high transpiration (Bonan et al., 2008; Forzieri et al., 2017). In fact, a net warming effect has been reported after tropical deforestation and agricultural expansion (Alkama et al., 2016; Duveiller et al., 2018). Furthermore, the biosphere...

• Line 18: In principle, it may serve also to detect where models work well. I would rephrase a bit the sentence in a more general way.

**Agree.**

In text: ... (CMIP5) models (Taylor et al., 2012; see Sect. 3.2 and 3.3). By comparing the observational and model-based results, areas where certain processes and inter-variable sensitivities may be incorrectly represented in ESMs, as well as others that match the observed behaviour, are identified.

• Line 31: Remote sensing LAI data in winter season are affected by snow cover conditions. I'm wondering how you have addressed this issue. If you did not account for this, I think your results may be strongly affected by this bias.

We agree with the reviewer that snow cover conditions might affect LAI in winter season, especially in high northern latitudes. However, we strongly believe that our results from CSGC, after adopting an ensemble for the observations and optimising the shape parameter, are trustworthy. For a full response to this question, we refer to comment #5.

**Page 4**

• Line 4: Why do you not use the ESA-CCI land cover product (and conversion to pass to PFT)? In principle, this enables to track for changes in PFT over almost the entire period of study. The ESA-CCI product represent the state-of-the-art product aimed to improve the link between remote sensing users and climate modellers... <a href="https://www.esa-landcover-cci.org/">https://www.esa-landcover-cci.org/</a>

We are aware of the ESA-CCI land cover product, and its possibilities. However, due to the fact that our current CSGC framework does not allow to detect changes over time in inter-variable sensitivities between climate and vegetation (see Reviewer #1 comment #3), we felt that for now, such a state-of-the art land cover product was overqualified. Also, the IGBP product is widely used, which simplifies comparison of our results with already published ones. Finally, as with all choices regarding data, the final choice remains subjective. However, we do see potential to use the ESA-CCI land cover product in our current work focussing in changes of explained variation between climate and the biosphere. We hope the reviewer agrees with our decision of using the IGBP product in this manuscript.

Line 8: Given the large differences amongst different products for some of the variables considered, I would strongly suggest to account for multiple products (https://onlinelibrary.wiley.com/doi/10.1111/gcb.13787; Jian et al., Inconsistencies of inter-annual variability and trends in long-term satellite leaf area index products, Global Change Biology, 23, 4133-4146, 2017). For instance, for LAI, data from GLASS, LTDR, GLOBMAP could also be included in the study. The same for precipitation which show large discrepancies – especially at inter-annual scale – depending on the dataset used. The use of ensemble of observational products would make your results more robust and substantially improve the work.

We strongly agree with the reviewer and adopted an ensemble for the observations. The manuscript and figures are updated accordingly. See comment #2 for more information.

Line 9: Please clarify the value of using online model simulations in place of offline simulations. I see a potential limitation as in online ESMs the climate signal may largely determine the response of the land surface and then mask the interplay between vegetation and biophysical processes. Further reading: Blyth et al., A comprehensive set of benchmark tests for a land surface model of simultaneous fluxes of water and carbon at both the global and seasonal scale, Geoscientific Model Development, 4(2), 255-269, 2011. (<a href="https://doi.org/10.5194/gmd-4-255-2011">https://doi.org/10.5194/gmd-4-255-2011</a>) and Winckler et al., Robust identification of local biogeophysical effects of land cover change in a global climate model, Journal of Climate, 30(3), 1159-1176. (<a href="https://doi.org/10.1175/JCLI-D-16-0067.1">https://doi.org/10.1175/JCLI-D-16-0067.1</a>)

We thank the author for this remark and for the articles of interest. Since CSGC is capable of unravelling interactions between variables without any assumption on the direction of these interactions, we saw the opportunity to use CSGC to investigate both the climate impacts on vegetation, and also the feedbacks of vegetation on climate. We also aimed to benchmark how these two-way interactions are represented in the ESMs, and therefore, we chose to work with online models as they do not only allow for climate to affect vegetation, but also for vegetation to provide a feedback on climate. In offline models, the separate parts of the models are driven by their necessary input, without any influence of the other parts. We added a brief description of this in the text.

In text: ... account for dynamic vegetation (Anav et al., 2015). Coupled model simulations are used to evaluate the full extent of vegetation feedbacks on climate. Using the historical...

• Line 13: Please, clarify the selection, why only these 4 models are used here?

These models are selected based on a combination of three criteria. All CMIP5 ESMs are selected based on the use of their land surface components also present in TRENDY initiative to allow for comparison with studies using these models, such as Forzieri et al. (2018). Also, only ESMs accounting for dynamic vegetation are retained and the models must provide hourly climate output (precipitation, air temperature and radiation). These criteria results in the selection of four models (CCSM4, HadGEM2-ES, NorESM1-M and IPSL-CM5A-MR). We added these criteria to the revised version of the manuscript.

In text: A selection of coupled ESMs from the Coupled Model Intercomparison Project Phase 5 (CMIP5; Taylor et al., 2012) is assessed in their representation of climate—vegetation interactions. This includes the Hadley Global... (CCSM4; Gent et al., 2011). This selection is based on (a) use of similar land surface schemes as the Trends in Net Land-Atmosphere Exchange (TRENDY; Sitch et al., 2015) initiative, in order to allow for comparison with studies focussing on TRENDY models, (b) availability of hourly input data for air temperature, precipitation, and net radiation (aggregated to monthly values in this study), and (c) model consideration of dynamic vegetation (Anav et al., 2015). Online model simulations...

• Line 15: Are all models run under a consistent modelling setup (e.g., same land cover changes, same climate forcing)? Please clarify. The consistency in modelling experiment is important to compare the different model results with each other

Yes, all models simulations where selected from the CMIP5 archive filtering for historical runs, ran with increasing greenhouse gas and sulphate aerosol concentrations, changes in solar radiation, forcing by volcanic aerosols, and time-evolving land-use changes (Anav et al., 2015).

 Line 17: Not sure this is correct. You basically used two different periods of analysis of observations and models: 1981-2015 (ca 35 years) for observations; 1956-2005 (50 years) for models. A part of the temporal shift between the two experiments, I would suggest to verify that the different length in the time series do not introduce a systematic bias between observational- and model-based results. Why you decided to start from 1956 for models? To me, it would be more logic at least to preserve the same length of observations (35 years). Please, check this and clarify your choices.

We agree that the change in period between observations and models may have an impact on our results. However, we analysed both observations and models for their overlapping period, and found no major changes to the results. The model results do faint for inter-annual scales due to the drastic decrease in sample points, but the general patterns remain, i.e. all regions show a similar climate control as for the 50-year period, albeit less clear due to the decreased sample size. Therefore, the models can be assumed to behave stationary and the period for models can be extended to 50 years to increase the robustness of the results. We added a brief description in the text and added the figures of the analysis for the overlapping period to the supplementary. For more information, we refer to the reply to reviewer #1, comment #6.

**Page 5**

• Line 24: In the present formulation of GC, the temporal lag m is implicitly assumed the same for all predictors. In practice, I expected that the legacy effects may differ depending on the predictor. Can this be included in the formulation? Please, discuss the implications.

To clarify, this formula refers to traditional Granger Causality, and not to the (Conditional) Spectral Granger Causality as calculated by Dhamala et al. (2008). CSGC, as proposed by Dhamala et al. (2008), is non-parametric and consequently no longer requires prescribing a specific lag; the dominant lag is in fact resolved by the formulation. For traditional Granger causality in the time domain, the parameter m represents the maximum temporal lag which best represents the dynamic of the system. This does not mean that all predictors have significant effect up to the lag m. A predictor might influence the system up to a lag k<m, which means that, for that predictor, the coefficients of the autoregressive model from k+1 to m will equal zero. So if the parameter m is chosen big, all possible legacy effects can be included in a multivariate case, and the most dominant lags will have the largest coefficients, and not necessarily at the same lag. However, by raising the maximum lag, also the computation costs increase. Therefore, a trade-off between included memory effects and computational cost needs to be taken into account with traditional Granger Causality. We hope resolves the remark. We also added a shorter version of this rationale to the manuscript.

In text: ...where m defines the maximum order of the autoregressive model (with  $m \le n$ ), i is the time lag,  $a_i$  are the coefficients describing the linear interaction between different time steps, and  $\epsilon_t$  is the prediction error. Note that the order m defines the maximum lag that is investigated, which does not necessarily imply that all predictors have an effect up to time step m. By increasing m, more lags are included, at the cost of increasing the computational demand.

• Line 27: In principle, data could be aggregated at seasonal or annual level and the GC applied to such values. I presume that the limited sample size hampers the use of GC in such an "aggregated" mode. Please clarify.

Yes, in principle, this is possible, and this is also roughly what Spectral Granger Causality does (but in a more elegant way). SGC decomposes the data in a time and frequency space using wavelet transformation. Then, instead of constructing the autoregressive models for each new time series at each determined frequency (which would require setting the parameter m arbitrarily, keeping in mind the computational costs), SGC is calculated the non-parametric way as described by Dhamala et al. (2008). Aggregating the data to seasonal and annual time series would create a limited sample size to determine GC, and would also generate abrupt breaking points in the data. Therefore, we apply Conditional Spectral Granger Causality instead of running the traditional Granger Causality in an aggregated mode. We shortly described this in the text as a justification to switch to Spectral Granger Causality.

In text: Despite traditional Granger causality being capable of addressing short-term interactions, simply aggregating time series to their seasonal or annual equivalents prior to following a traditional Granger causality approach does not necessarily lead to realistic causation inference at larger temporal scales. Consequently, Granger causality frameworks that are defined in the time domain...

**Page 6**

• Line 2: ...assess temporal scale-dependent...

**Adapted in text.**

• Line 19: I believe that Eq. 7 needs more clarification for readers not familiar with the method.

We thank the reviewer for the concern and agree that a reader unfamiliar with the method needs more clarification. The method starts with traditional Granger Causality, followed by a description of Spectral Granger Causality, and ends with Conditional Spectral Granger Causality. We feel that further explanation of SGC won't benefit the reader, but refer to, in our opinion, excellent introductory material on the method and its application to ecological problems. Readers that are truly interested in understanding the method are better suited in studying this material rather than trying to understand it from the limited space we have available in this manuscript.

In text: ... using matrix factorisation (Wilson, 1972). For more information on SGC, we refer to Ding et al. (2006), Dhamala et al. (2008) and Detto et al. (2012; 2013).

• Line 29: Zp in place of Xp

A minor mistake did indeed slip into the formula. We corrected for it. We thank the reviewer for noticing it.

 Line 14: I would clearly mention that CSGC does not allow to quantify the sign of causal relation. It is already mentioned in results... but I would also mention here – or somewhere in the method section – because important.

We agree with the reviewer and added a sentence stating that CSGC can only detect that there is a relation between variables, but not the sign of the causal relation. However, we also like to clarify that the sign of causality is less meaningful in time series analyses, where the concept of positive and negative interactions is substituted by the phase angle. For example, a positive interaction between time series corresponds to a zero phase angle (the series are in phase or perfectly synchronised, i.e. when one goes up, to other goes up as well). Conversely, a negative interaction implies that the series are out of phase (one goes up, the other goes down). These series would have an arbitrary angle of 180° or -180°. These phase angles can be determined using co-spectral analyses, but not directly from the Granger causality measure.

In text: ... However, if there is a direct causal influence of X on Y at a specific frequency f,  $CSGC_{X \to Y|Z_1,Z_2...Z_p}(f) > 0$ . Using Eq. 10, it is possible to determine if X (Granger-) causes Y, but no information on the sign of the causal relation can be extracted.

**Page 9**

• Line 3: I would refer to seasonal LAI variability here and in the rest of the manuscript. Phenology implies other metrics that are not accounted for in this work.

We sincerely thank the author for the concern. We hope our reply to the comment on page 1, line 9 clears the confusion.

• Line 8: Please clarify this upper value.

Theoretically,  $+\infty$  is the largest timescale, calculated using extrapolation. However, the first real maximum value equals 16.5 years due to the discretisation of the frequency phase by CSGC. The frequency phase is discretised in angular frequency from 0 to  $\pi$ , in 100 values (default). As frequency [1/month] equals the angular frequency divided by  $2\pi$ , and since scale [years] equals 1/(Frequency\*12), the maximum scale (corresponding to an angular frequency of  $\pi$ /99) equals 16.5 years. By tweaking the discretisation level, the maximum real scale could be raised to exactly 17 years (half of the total period for observations). However, since 16.5 years is extremely close to 17 years, and as we do not study the inter-decadal patterns, we opted for using the default value of 100 discrete frequencies, evenly spaced throughout the angular frequency space. We added a brief description to the method section.

In text: ...between climate and vegetation. Moreover, based on the characteristics of the climate data used in this study, CSGC can be applied to assess causality over a wide range of temporal scales, starting at 2 months (twice the temporal resolution) and going up to 16.5 years (maximum temporal scale due to discretisation of the frequency space; can be adjusted if needed, especially for longer time series).

**Page 10**

• Line 9: Could irrigation or land enlargement, particularly relevant in some regions of the globe, partially explain some patterns (e.g. India and China). Should the irrigated lands not be factored out?

https://www.nature.com/articles/s41893-019-0220-7 (Chen et al., China and India lead in greening of the world through land-use management, Nature Sustainability, 2, 122-129, 2019)

We thank the reviewer for this comment. We could factor out irrigated lands and other disturbed surfaces, but chose to only exclude non-vegetated areas such as desserts. We acknowledge that practices such as irrigation can impact the results. Here, only three dominant climate drivers are addressed, meaning that if human activities, such as e.g. irrigation, are the main driver of vegetation in an area, as could be in India and China, these regions probably get an energy-related driver attributed to them. However, if maps would be created of the part of variance that is not explained by the (considered) climate variables, areas with high human impact would be highlighted (see figure below, showing the variability in LAI not driven by climate). As can be seen, areas such as India and China show close to 100% of variation in LAI not driven by climate (white colors) at monthly scales. We chose not to show these maps in the supplementary, to avoid an abundance of plots. However, if the reviewer finds it necessary to include these maps, we can add them. In the manuscript, we chose not to exclude these areas, as also irrigated lands remain partly driven by climate, albeit to a limited extent. We added a paragraph to the manuscript to discuss this issue (see reviewer #1 comment #4)

---

## Referee Report (RR1)

**Review "Global biosphere-climate interaction: a multi-scale appraisal of observations and models by J. Claessen"**

G. Forzieri

I would like to thank the authors for their extra-efforts in order to account my major concerns. I have sincerely appreciated their work. I have still a series of comments. The most important ones relate to the need of additional clarifications in the text, issues related to the significance and some possible numerical inconsistency amongst figures.

**Page 1**

Line 10: I find this a bit redundant ... would not be better to refer explicitly to what geographic regions they refer?

Line 12: I would explicit four models, as now it seems that you tested the whole CMIP ensemble

Line 15: I would try to be consistent here using "vegetation". You are not focusing only on forests...

Line 18: I think this is an over-interpretation. I would remove the downwind influence.

**Page 2**

Line 28: I would suggest to split this long sentence.

Line 26: explicit four models

Line 27: I would suggest to restructure a bit this sentence.

**Page 4**

Table 1:I would suggest to distinguish this info in t fields, as: spatial resolution, temporal resolution, temporal coverage

Line 8: I would explicit in the text the temporal coverage so that the info provided in the following section can be better understood

**Page 9**

Line 8: CSGC, ... add comma Line 16: typo: remove "is" Line 29: Would it make sense to refer to figure 1b?

**Page 10**

Line 34: Refer to fig.1a?

**Page 11**

Line 3: I would add in a dedicated section or somewhere in the methods a description on the multiproduct (or multi-model) ensembles. Describing for instance how you utilized the different observational products (modes). For observations, I suppose that you run the CSGC for each combination of LAI, T, P, Radiation and then compute the average over the ensemble of results. the model ensemble has 4 members. What about the size of the observational ensemble? Please, clarify and provide this info in the text.

Line 15: I have still some concerns about this issue. To my understanding, some LAI products have a lot of missing data during winter periods. It would be important that authors clarify how they solved this issue. Times with missing data are simply excluded the time series? Do this affect the time-frequency transformation? Did you keep pixels with a minimum number of observations during winter seasons? This details should be clarified maybe in the method section.

Line 22: not need of parenthesis, just put a comma.

Line 24: Looking at figure 2a, I would say that radiation is the primary control on a larger portion of vegetated land... can you please check these numbers? I suppose, you have accounted for the dependence of the spatial extent of grid cells on the latitude.

Line 30: I would restructure a bit this sentence...this is clearly in response to my previous comment, but as now you are assuming the reader is complaining about the effect of irrigation...

**Page 13**

Line 5: I agree with this interpretation. It would be interesting to check the climate impact on vegetation for different irrigation patterns. For instance, plotting separately P, Ta, Rn controls over a gradient of percentage of irrigated area. I would expect that an increase irrigation, for some areas, may tend to amplify the control of Rn and Ta (as also argued by authors). The analysis should be relatively easy to perform and could add interesting material to your interpretation.

A possible dataset to explore this:

http://www.fao.org/nr/water/aquastat/irrigationmap/index.stm

It is more complex to extend the same rationale to explore the effect of deforestation. One option, not sure about the results, could be to compute the long-term linear trends in cover fraction of non-forest classes (including natural grasses, crops, bare soil), for instance from annual land cover maps. The trend computed at the 0.5 degree spatial resolution should reflect the long-term transition from forest to non-forest state. The idea would be to dependence of the climate controls across a gradient of deforestation rate (the retrieved linear trend).

Possible datasets to explore this:

https://www.esa-landcover-cci.org/

https://earthenginepartners.appspot.com/science-2013-global-forest/download v1.2.html

https://www.nature.com/articles/s41586-018-0411-9? sg=RshFR75ayixme4g7CasqNzvfZF2Mw0bNFQFtRQcDkFh7 rLj96NyYt0O3442MI74umt9nXWfDbnR A.bLXfW RzdtPGLPmYQ16GWH9fCT3jhEyrtMsymQU fr1sGhDI 4iXJiEbqMaOxTXxm-3NyW9qnZ-7WwWgJfw9BkQ

Analogously, the same could be computed for the vegetation impact on climate (for observation and models). In theory could be described in a dedicated section to better disentangle the human effect. Please, consider this as a suggestion and develop only you think that can add important material. As potential reader of your work, I believe that this analysis would add value to your analysis.

Line 30: energy-to-water?

Line 32: why? Please, clarify

**Page 15**

Line 14: Can you please comment the low significance of these results? Seems that the inter-product spread tend to substantially alter the significance level (compare with your results in the previous submission). Are different combination of observational products leading to completely different results? To me it is a bit worrying such low level of agreement amongst the products. It may also be a warning in the use of such product (with results so different each other) to benchmark ESMs. I would really appreciate some more details on this.

Line 21: There is also a substantial overestimation of the LAI control on Ta and an underestimation of LAI control on PP in models compared to observations at monthly scale (see for instance sub-tropics). Am I right? Can you please comment on this?

**Page 16**

Line 14: due to compensatory effects?

Line 22: Please, clarify this statement. Without an appropriate context this appears a pure speculation. Based on the same rationale, one could say that also shadow effects, or many other micro-biometereological effects, are not accounted for...

I honestly think that you do not have enough data to support this statement.

Line 33: global in place of continental?

**Page 17**

Line 8: This appears in figure 3 but not in figure 6. To me there are some inconsistencies between latitudinal profiles in figure 3 and the global averages shown in figure 6. Just as example, compare Fig. 6b with Fig. 3a. The average absolute vegetation feedback on radiation is about 43% in Fig. 6b. If you look at the latitudinal profile in Fig. 3a, values of vegetation feedback on radiation are always below 10%. Please, check these numbers.

Code availability: please, be sure to have uploaded your codes, or to have properly labeled them. I had a look, and I could not find the codes related to this publication.

**Figure 1**

if in the text you refer to panel b earlier than panel a, I would suggest to change the order of the panels to be consistent with the text

**Figure 2**

- Not sure if this can be easily visualized, but it would be interesting to see the inter-product and inter-model spread. It would be interesting to compare the uncertainty across observational products and that one emerging from models... If the information is relevant you could consider to move the panels with latitudinal profiles in a separate figure in order to better visualize results.
- I would suggest to use a different color or style for the reticular grid of the maps. Now, it tend to be confused with black dots of significance. Same for the other figures
- I would suggest to put at the corners of the triangle the number "100%" in addition to the variable label. For instance "Ta 100%". the same for the other figures.

**Figure 6**

I would say just global averages... There is no need to introduce such new regional aggregation term. You do not distinguish values amongst continents.

---

## Author Response (AR2)

We sincerely thank Dr. Forzieri for the comments on the manuscript. Beneath, each concern is tackled separately. First the comments is listed, followed by our reply in italics. Also, if the manuscript was changed accordingly, this is added to our reply (underlined sentences represent the alterations to the draft).

We also like to thank the anonymous reviewer for his acceptance of the manuscript.

Directly after the responses to all comments, we include a marked-up version of the manuscript to track the changes.

**Reply to the comments by Dr. Forzieri**

- I would like to thank the authors for their extra-efforts in order to account my major concerns. I have sincerely appreciated their work. I have still a series of comments. The most important ones relate to the need of additional clarifications in the text, issues related to the significance and some possible numerical inconsistency amongst figures.

*We thank the reviewer for his appreciation of the revised manuscript, but even more for the suggestions made in the previous review round.*

**Page 1**

- Line 10: I find this a bit redundant… Would not be better to refer explicitly to what geographic regions they refer.

*We decided not to change the statement to avoid repetition of "northern latitudes" in the same sentence.*

- Line 12: I would explicit four models, as now it seems that you tested the whole CMIP ensemble.

*We thank the reviewer for his suggestion and altered the text accordingly.*

*In text: …as benchmark to evaluate four ESM simulations*

- Line 15: I would try to be consistent here using "vegetation". You are not focussing only on forests…

*We referred here to forest variability as Fig. 4 clearly showed that the air temperature control was overestimated by ESMs in forest biomes. However, we altered the statement to refer to vegetation in general.*

*In text: …control of air temperature on seasonal vegetation variability, especially in forested regions.*

- Line 18: I think this is an over-interpretation. I would remove the downwind influence.

*We do not fully agree, as we explain in Sect. 3 how to local character of our analysis can impact our results. However, we do agree that this statements needs clarification.*

*In text: …Note however that while vegetation reacts directly to its local climate conditions, the spatially collocated character of the analysis does not allow for the identification of remote feedbacks, which might result in an underestimation of the biophysical effects of vegetation on climate. Nonetheless, the…*

**Page 2**

- Line 28: I would suggest to split this long sentence.

*We chose to rephrase and condense the sentence.*

*In text: Likewise, observational data have been used to benchmark vegetation variability in ESM's (Anav et al., 2013, Murray-Tortarolo et al., 2013), and an overestimation of modelled annual LAI due to problems related to the timing of the phenological cycle has been suggested (Forkel et al., 2014; Verger et al., 2016). Rather…*

**Page 3**

- Line 26: explicit four models.

*Agreed.*

*In text: …from four online simulations…*

- Line 27: I would suggest to restructure a bit this sentence.

*Agreed.*

*In text: …By comparing the observational and model-based results, areas with matching or diverging inter-variable sensitivities are identified.*

**Page 4**

- Table 1: I would suggest to distinguish this info in fields, as: spatial resolution, temporal resolution, temporal coverage.

*Agreed.*

*In text:*

| Product | Variable | Spatial Resolution | Temporal Resolution | Temporal Coverage | Reference |
|---|---|---|---|---|---|
| Global Inventory Modelling and Mapping Studies 3re generation (GIMMS3g) | LAI | 1/12° | Bimonthly | 1982-2015 | Zhu et al. (2013) |
| NOAA/AVHRR Thematic Climate Data Record (TCDR) Reflectance | LAI | 0.05° | Daily | 1982-2018 | Claverie et al. (2016) |
| GIMMS3g + Terra/MODIS C5 reflectance (GLOBMAP) | LAI | 1/13.75° | 28-day | 1982-2017 | Liu et al. (2012) |
| NOAA/AVHRR LTDR + Terra/MODIS C5 reflectance (GLASS) | LAI | 0.05° | 8-day | 1982-2015 | Xiao et al. (2016) |
| European Centre for Medium-Range Weather Forecasts (ECMWF) ERA5 | Ta, Rn and P | 32km | Hourly | 1979-… | Hersbach and Dee (2016) |
| Climate Research Unit – National Centers for Environmental Prediction (CRU-NCEP) version 7 | Ta, Rn and P | 0.05° | 6-hour | 1901-2016 | Viovy (2018) |
| Global Precipitation Climatology Centre (GPCC) | P | 0.5° | daily | 1891-2016 | Schneider et al. (2011) |

- Line 8: I would explicit in the text the temporal coverage so that the info provided in the following section can be better understood.

*Agreed.*

*In text: …Tab. 1 provides an overview of the available data sets, resulting in the overlapping analysis period 1982—2015.*

**Page 9**

- Line 8: CSGC, add comma

*Agreed.*

- Line 16: typo: remove "is"

*We thank the reviewer for noticing this typo.*

- Line 29: Would it make sense to refer to figure 1b?

*We thank the author for his suggestion and included a reference to Fig. 1b in the manuscript (Fig. 1a in updated manuscript).*

*In text: …an intuitive manner (see Fig. 1a and Sect. 3).*

**Page 10**

- Line 34: Refer to Fig. 1a?

*Agreed. (Referred to Fig. 1b due to updated Fig. 1).*

*In text: …See Fig. 1b for an…*

**Page 11**

- Line 3: I would add in a dedicated section or somewhere in the methods a description on the multi-product (or multi-model) ensembles. Describing for instance how you utilised the different observational products (modes). For observations, I suppose that you run the CSGC for each combination of LAI, T, P, Radiation and then compute the average over the ensemble of results. The model ensemble has 4 members. What about the size of the observational ensemble? Please, clarify and provide this info in the text.

*We agree with the reviewer that a description of the ensembles is lacking. The model-ensemble has indeed 4 members, and a 48-member ensemble for the observations (4LAI x 3P x 2Ta x 2Rn). The ensemble results are provided by averaging (or determining the specific quantiles) over all members. We added a brief description in the sections addressing the observational and models.*

*In text, in Sect. 2.1.1.: …Tab. 1 provides an overview of the available data sets, resulting in the overlapping analysis period 1982—2015. The main observational results are based on the average of the 48-member ensemble, acquired by analysing all possible data set combinations.*

*In text, in Sect. 2.1.2.: …divergences between the observation and model results. The latter will be presented as the average over the four model ensemble members.*

- Line 15: I still have some concerns about this issue. To my understanding, some LAI products have a lot of missing data during winter products. It would be important that authors clarify how they solved this issue. Time with missing data are simply excluded the time series? Do this affect the time-frequency transformation? Did you keep pixels with a minimum number of observations during winter seasons? This details should be clarified maybe in the method section.

*We can relate to the concern of the reviewer. LAI products are known the contain artefacts, especially in winter time. Gaps in the products (of both LAI and climate), are resolved by a spatial bilinear interpolation. However, this does not resolve pixels contamination by snow cover. Therefore, after the suggestion of the reviewer in the previous revision round, we opted for using an ensemble of products to cancel out product-specific artefacts. Excluding data outside the growing season from the time series isn't possible with CSGC as described here, but might be an option with the time-defined CSGC, which is out of the scope of this paper. We hope the reviewer agrees with our rationale, which we briefly added to the manuscript.*

*In text, in Sect. 2.1.1.: …these variables (Jiang et al., 2017; Sun et al., 2018). LAI products have data gaps and higher uncertainties in winter periods (Yang et al., 2006; Xiao et al., 2016; Jiang et al., 2017). Gaps are here filled by bilinear interpolation as CSGC requires continuous time series. Tab. 1…*

- Line 22: not need of parenthesis, just put a comma.

Agreed.

- Line 24: looking at figure 2a, I would say that radiation is the primary control on a larger portion of vegetated land… Can you please check these numbers? I suppose, you have accounted for the dependence of the spatial extent of the grid cells on the latitude.

*We thank the reviewer for his concerns. The percentages given in the manuscript already accounted for the pixel area size, which explains the apparent contradiction with the maps.*

- Line 30: I would restructure a bit this sentence… This is clearly in response to my previous comment, but as now you are assuming the reader is complaining about the effect of irrigation…

*Agreed. The statement is altered to first introduce that irrigation might affect our results.*

*In text: …precipitation only in this study. Also, human practices, such as i.e. irrigation, can potentially bias our results. Nonetheless, irrigation is expected to increase the energy-dependency of LAI dynamics, and as irrigation tends to be a seasonal phenomenon restricted to the growing period, this increase is found to be clearer as seasonal than monthly scales (as shown in Supplementary Fig. B1).*

**Page 13**

- Line 5: I agree with this interpretation. It would be interesting to check the climate impact on vegetation for different irrigation patterns. For instance, plotting separately P, Ta, Rn controls over a gradient of percentage of irrigated area. I would suspect that an increase in irrigation, for some areas, may tend to amplify the control of Rn and Ta (as also argued by authors). The analysis should be relatively easy to perform and could add interesting material to your interpretation. A possible dataset to explore this: http://www.fao.org/aquastat/en/geospatial-information/global-maps-irrigated-areas/latest-version/ It is more complex to extend the same rationale to explore the effect of deforestation. One option, not sure about the results, could be to compute the long-term linear trends in cover fraction of non-forest classes (including natural grasses, crops, bare soil), for instance from annual land cover maps. The trend computed at the 0.5 degree spatial resolution should reflect the long-term transition form forest to non-forest state. The idea would be to dependence of the climate controls across a gradient of deforestation rate (the retrieved linear trend). Possible datasets to explore this: https://www.esa-landcover-cci.org/ , https://earthenginepartners.appspot.com/science-2013-global-forest/download_v1.2.html , https://www.nature.com/articles/s41586-018-0411-9?_sg=RshFR75ayixme4g7CasqNzvfZF2Mw0bNFQFtRQcDkFh7_rL-j96NyYt0O3442Ml74umt9nXWfDbnR_A.bLXfW_RzdtPGLPmYQ16GWH9fCT3jhEyrtMsymQU_fr1sGhDI4iXJiEbqMaOxTXxm-3NyW9qnZ-7WwWgJfw9BkQ Analogously, the same could be computed for the vegetation impact on climate (for observation and models). In theory could be described in a dedicated section to better disentangle the human effect. Please consider this as a suggestion and develop only you think that can add important material. As potential reader of your work, I believe that this analysis would add value to your analysis.

*We thank the reviewer for the suggestions. Below, we inserted the suggested figure, showing the climate control as a function of the percentage of area equipped for irrigation. As the highest levels of irrigation occur in monsoonal regions (India and China), irrigation itself is a highly seasonal practice. Therefore, we can expect to see a decrease in precipitation control at seasonal scales for high levels of irrigation, accompanied by an increase of the energy-dependency. Despite this confirmation, we believe that to fully unravel the effects of irrigation, we need a method that is capable to exclude periods without irrigation. This would require a time-based version of CSGC, which is out of the scope of this manuscript, but is a current work in progress. Therefore, we opted to include this figure as a supplement to the article, and briefly refer to it in the main text (see previous comment).*

*The question of deforestation is harder to tackle, and definitely needs a time-defined version of CSGC. We agree that it is an important question that needs to be addressed, but feel that our implementation of CSGC is not capable of correctly tackling it. We hope to reviewer agrees with this rationale*

[Figure]

- Line 30: energy-to-water?

*To be consistent with the region we define as south-to-north in the Amazonia, the gradient in climate controls must be water-to-energy.*

- Line 32: why? Please, clarify

*We added a brief explanation stating that only air temperature is found to be a limiting climate control.*

*In text: …These differences might indicate difficulties to model climate-vegetation interactions across the basin where air temperature is found to be the only limiting control, yet they may again be influenced…*

**Page 15**

- Line 14: Can you please comment the low significance of these results? Seems that the inter-product spread tend to substantially alter the significance level (compare with your results in the previous submission). Are different combination of observational products leading to completely different results? To me it is a bit worrying such low level of agreement amongst the products. It may also be a warning in the use of such product (with results so different each other) to benchmark ESMs. I would really appreciate some more detail on this.

*The observational results did lose some significant patterns due to adopting an ensemble approach, especially for the feedbacks of vegetation on climate. The reason can be two-fold. First of all, the magnitude of the feedbacks is less than that of the climate impact, which lowers the amount of dots for the feedbacks. This is also apparent in the figures of our first submission. Secondly, due to adopting an ensemble, we increased the homogeneity of the average patterns, but due to some heterogeneity over ensemble members, the average of the ensemble is not found to be significant. We briefly added this to the text.*

*In text: …the somehow limited observational record. Individual ensemble members do achieve high significance, but little inter-product agreement is reached due to high spatial heterogeneity over ensemble members. Overall, and as expected…*

- Line 21: There is also a substantial overestimation of the LAI control on Ta and an underestimation of LAI control on P in models compared to observations at monthly scale (see for instance sub-tropics). Am I right? Can you please comment on this?

*We agree with the reviewer that this effect is not limited to tropical forests. We altered the manuscript.*

*In text: …significant impacts of LAI on precipitation in the (sub-)tropics, these effects are …*

**Page 16**

- Line 14: due to compensatory effects?

*Yes. We added this to the manuscript.*

*In text: …looked at biome-averaged patterns due to compensatory effects, as comparison…*

- Line 22: Please clarify this statement, without an appropriate context this appears pure speculation. Based on the same rationale, one could say that also shadow effects, or many other micro-biometeorological effects, are not accounted for… I honestly think that you do not have enough data to support this statement.

*We thank the reviewer for his comment and agree that the statement is missing a proper context. We altered the statement accordingly.*

*In text: …non-consideration of downwind influences in this analysis which have been shown to be crucial (Dirmeyer et al., 2009; Zemp et al., 2017). However, similar…*

- Line 33: global in place of continental?

*Agreed but resolved alternatively.*

*In text: …Averaged over all vegetated land, radiation…*

**Page 17**

- This appears in figure 3 but not in figure 6. To me there are some inconsistencies between latitudinal profiles in figure 3 and the global averages shown in figure 6. Just as example, compare Fig. 6b with Fig. 3a. The average absolute vegetation feedback on radiation is about 43% in Fig. 6b. If you look at the latitudinal profile in Fig. 3a, values of vegetation feedback on radiation are always below 10%. Please, check these numbers.

*We thank the reviewer for noticing this inconsistency. Fig. 6 showed the percentage of land over which a specific driver (or feedback) dominated, a not the absolute value averaged over the globe. Fig. 6 is updated, and some statements were altered to be in line with Fig. 6.*

*In text: …but models seem incapable of reproducing the observed spread in the strength of this dependency. ESMs generally overestimate the precipitation control on vegetation, and most drastically at inter-annual scales. …while feedbacks on local precipitation are often underestimated, especially at seasonal and inter-annual time scales. Finally, interactions…*

- Code availability: please, be sure to have uploaded your codes, or to have properly labelled them. I had a look, and I could not find the codes related to this publication.

*The codes still need to be uploaded to our GitHub-page, which will be performed at the time of publication.*

***Figure 1***

- If in the text you refer to panel b earlier than panel a, I would suggest to change the order of the panels to be consistent with the text.

*Fig. 1 is updated in the manuscript.*

**Figure 2**

- Not sure if this can be easily visualised, but it would be interesting to see the inter-product and inter-model spread. It would be interesting to compare the uncertainty across observational products and that one emerging from models… If the information is relevant, you could consider to mover the panels with latitudinal profile in a separate figure in order to better visualise results.

*We thank the author for the suggestion. We created figures showing the coefficient of variation for the climate impact on vegetation for both observations and models. The average coefficient of variation over all three climate drivers is shown. These figures don't seem to add value to explain differences between observations and models, as there are no consistent patterns between the coefficient of variation and significance of the results. Therefore, we chose not to add the maps to the manuscript.*

[Figure]

- I would suggest to use a different color or style for the reticular grid of the maps. Now, it tend to be confused with black dots of significance. Same for the other figures.

*Agreed. Fig. 2, Fig. 3 and all relevant additional figures are updated.*

- I would suggest to put at the corners of the triangle the number "100%" in addition to the variable label. For instance "Ta 100%". The same for the other figures.

*Agreed. Fig. 2, Fig. 3 and all relevant additional figures are updated.*

**Figure 6**

- I would say just global averages… There is no need to introduce such new regional aggregation term. You do not distinguish values amongst continents.

*Agreed. The caption is updated.*

*In text: Global average … Global averages…*

[revised manuscript text omitted]